# Common cell type nomenclature for the mammalian brain

Jeremy A Miller*, Nathan W Gouwens, Bosiljka Tasic, Forrest Collman, Cindy TJ van Velthoven, Trygve E Bakken, Michael J Hawrylycz, Hongkui Zeng, Ed S Lein, Amy Bernard*

Allen Institute, Seattle, United States

**Abstract** The advancement of single-cell RNA-sequencing technologies has led to an explosion of cell type definitions across multiple organs and organisms. While standards for data and metadata intake are arising, organization of cell types has largely been left to individual investigators, resulting in widely varying nomenclature and limited alignment between taxonomies. To facilitate cross-dataset comparison, the Allen Institute created the common cell type nomenclature (CCN) for matching and tracking cell types across studies that is qualitatively similar to gene transcript management across different genome builds. The CCN can be readily applied to new or established taxonomies and was applied herein to diverse cell type datasets derived from multiple quantifiable modalities. The CCN facilitates assigning accurate yet flexible cell type names in the mammalian cortex as a step toward community-wide efforts to organize multi-source, data-driven information related to cell type taxonomies from any organism.

*For correspondence:
jeremym@alleninstitute.org (JAM);
amyb@alleninstitute.org (AB)

**Competing interests:** The authors declare that no competing interests exist.

## Introduction

Cell type classification has been central to understanding biological systems for many tissues (e.g., immune system) (*Lees et al., 2015*) and organisms (e.g., *Caenorhabditis elegans*) (*Packer et al., 2019*). Identifying and naming cellular components of the brain has been an integral part of neuroscience since the seminal work of *Cajal, 1899*. Many neuronal cell types, such as neurogliaform, chandelier, Martinotti, and pyramidal cells, have been identified based on highly distinct shape, location, or electrical properties, providing robust and consistent classifications of neuronal cell types and a common vocabulary (*Greig et al., 2013*; *Markram et al., 2004*). However, the recent application of high-throughput, quantitative methods such as single-cell or -nucleus transcriptomics (scRNA-seq) (*Hodge et al., 2019*; *Macosko et al., 2015*; *Saunders et al., 2018*; *Tasic et al., 2018*; *Tasic et al., 2016*; *Zeisel et al., 2018*; *Zeisel et al., 2015*), electron microscopy (*Zheng et al., 2018*), and whole brain morphology (*Winnubst et al., 2019*) to cell type classification is enabling more quantitative measurements of similarities among cells and construction of taxonomies (*Zeng and Sanes, 2017*). The use of scRNA-seq, in particular, for cell type classification has increased exponentially since its introduction a decade ago (*Tang et al., 2009*), with nearly 2000 published studies and several hundred tools for data analysis (*Zappia et al., 2018*). These methodological advances are ushering a new era of data-driven classification, by simultaneously expanding the number of measurable features per cell, the number of cells per study, the number of classification studies, and the computational resources required for storing and analyzing this information.

This data explosion has enriched our collective understanding of biological cell types, while simultaneously introducing challenges in cell type classification within individual studies. In the retina, neurons with shared morphology also have consistentconnectivity (*Jonas and Kording, 2015*), spacing, arbor density, arbor stratification (*Seung and Sümbül, 2014*), and gene expression signatures (*Macosko et al., 2015*), often with one-to-one correspondences between phenotype and function (*Zeng and Sanes, 2017*). However, studies combining scRNA-seq with traditional morphological and

electrophysiological characterizations in the brain have found a more complicated relationship in the brain than in retina, with cell types defined by morphology and electrophysiology sometimes containing cells from several cell types defined using gene expression (*Gouwens et al., 2020*; *Kozareva et al., 2020*), and some transcriptomically defined types containing cells with multiple morphologies (*Hodge et al., 2020*; *Hodge et al., 2019*). Further complicating classification is the overlay of discrete cell type distinctions with graded/continuous properties such as cortical depth (*Berg et al., 2020*), anterior/posterior and other trajectories across neocortex (*Hawrylycz et al., 2012*), activity-dependent cell state (*Wu et al., 2017*), or all simultaneously (*Yao et al., 2020b*). Furthermore, functional properties observed in matched cell types may diverge across species (*Bakken et al., 2020a*; *Berg et al., 2020*; *Boldog et al., 2018*; *Hodge et al., 2019*) and as cells advance along trajectories of development (*Nowakowski et al., 2017*), aging (*Tabula Muris Consortium, 2020*), and disease (*Mathys et al., 2019*).

Given this complex landscape, determining fundamental criteria for cell type definition in a given study, and then establishing correspondence to a cell type defined in another study, is often nontrivial and sometimes impossible. Substantial progress has been made toward solving this challenge of 'alignment', whereby datasets collected with genomics assays such as scRNA-seq or snATAC-seq can be used to anchor diverse cell types in a common analysis space (*Barkas et al., 2018*; *Butler et al., 2018*; *Johansen and Quon, 2019*). Alignment has proven effective for matching cell type sequence data collected on different platforms, across multiple data modalities, and even between species where few homologous marker genes show conserved patterns (*Bakken et al., 2020a*; *Bakken et al., 2020b*; *Hodge et al., 2020*; *Hodge et al., 2019*; *Yao et al., 2020a*). When combined with experimental methods such as Patch-seq (*Cadwell et al., 2016*; *Fuzik et al., 2016*; *Gouwens et al., 2020*; *Scala et al., 2020*), which involves application of electrophysiological recording and morphological analysis of single patch-clamped neurons followed by scRNA-seq of cell contents, autoencoder-based dimensionality reduction (*Gala et al., 2019*) can extend these alignments to bridge distinct modalities. Such analysis strategies provide a mechanism for classifying cell types using data from disparate data sources, allow for annotation transfer between experiments, and are a critical step toward unifying data-driven cell type definitions. However, as new cell type classifications are continually emerging, it is unrealistic to expect complete alignment of all published datasets, but creation of standardized systems for alignment becomes even more important.

Standardized cell type classification needs to include (1) standard nomenclature and (2) centralized and standardized infrastructure associated with cell type classification. Such standards provide a mechanism for storing key information about cell types and assigning explicit links between common cell types identified in different studies. Currently, no standard convention of naming brain cell types is widely followed. Cell types have historically been named by their shape, location, electrical properties, selective neurochemical markers, or even the scientist who discovered them (*Betz, 1874*; *Szentágothai and Arbib, 1974*). Now, quantitative clusters that cannot obviously be matched with these types are named on an ad hoc basis, either by assigning generic names like 'interneuron 1' or 'Ex1' and then linking these names to associated figures, tables, or text (*Gouwens et al., 2019*; *Lake et al., 2016*; *Zeisel et al., 2015*), or by chaining critical cell type features in the name itself, resulting in names like 'Neocortex M1 L6 CT pyramidal, *Zfpm2* non-adapt GLU' (*Shepherd et al., 2019*). All of these proposals are reasonable for stand-alone projects but make direct comparisons between studies daunting. While several public databases for data storage have been developed (e. g., dbGaP, NeMO, NeuroElectro, Neuromorpho, HuBMAP, etc.), a community-recognized repository for storing and tracking cell type assignments and associated taxonomies does not currently exist. This challenge has been recognized by many (*Armañanzas and Ascoli, 2015*; *DeFelipe et al., 2013*; *Shepherd et al., 2019*) and has been a focus of recent conferences seeking community participation toward a solution (*Yuste et al., 2020*). Any solution devised to tackle this question should ideally be effective and user-friendly and should directly address some of the ongoing challenges of ontology, data matching, and cell type naming described above in its implementation, providing some amount of immediate standardization of any cell type classifications included therein. This challenge was also addressed at a *Cell Type Ontology Workshop* (Seattle, June 17–18, 2019; hosted by the Allen Institute, Chan Zuckerberg Initiative [CZI] and the National Institutes of Health [NIH]), where input from representatives from the fields of ontology, taxonomy, and neuroscience made recommendations, highlighted best practices, and proposed conventions for naming cell types.

To begin to address these challenges and driven by a practical need to organize vast amounts of multimodal data generated by the Allen Institute and collaborators, we have developed a nomenclature convention aimed at tracking cell type information across multiple datasets. Here we present a generalizable nomenclature convention, the **common cell type nomenclature** (CCN), for matching and tracking cell types across studies. This convention was motivated by methodologies used for management of gene transcript identity tracked across different versions of GENCODE genome builds, allowing comparison of matched types with a common reference or any other taxonomy (*Frankish et al., 2019*; *Harrow et al., 2012*). Motivated by gene nomenclature conventions from HGNC (*Bruford et al., 2020*), the CCN also facilitates assigning accurate yet flexible cell type names in the mammalian cortex as a step toward community-wide efforts to organize multi-source, data-driven information related to cell type taxonomies from any organism. An initial version of the CCN was introduced at https://portal.brain-map.org/explore/classes (October 2019), with the intent to encourage discussion and gather feedback for improving subsequent versions, to facilitate collaboration, and to improve shared understanding of the many cell types in the brain.

## Results

### Overview of proposed nomenclature convention

The problem of defining and naming cell types has many similarities to those of genes in genomics, where there is a practical need to track individual sequencing and assembly results as distinct and self-contained entities, while simultaneously recognizing the goal for a singular reference that the community can use to map sequencing data into a common context (*Frankish et al., 2019*; *Harrow et al., 2012*; *Kitts et al., 2016*). Here, a similar strategy is proposed for cell type nomenclature: Use of a standardized series of identifiers for tracking cell types referenced to individual studies, in addition to providing a mechanism for defining common identifiers (*Figure 1A*). At the core of the schema are two key concepts: (1) a **taxonomy**, defined as the output of a computational algorithm applied to a specific **dataset**, which must be generated prior to implementation of this schema, and (2) a **cell set**, which can represent any collection of **cells** within a specific taxonomy (see *Table 1* for definitions of key terms). These components are generated through the input of data and information generated from analysis that identifies **provisional cell types** (sometimes called **cell types** for convenience). These are analytically relevant cell sets that represent quantitatively derived data clusters defined by whatever classification algorithm generated the taxonomy. Provisional cell types can be organized as the terminal leaves of a hierarchical taxonomy using a **dendrogram**, as a non-hierarchical **community structure**, or both. Taxonomies and cell sets are assigned unique identifier tags, as described below, and additional metadata can be stored alongside these tags for use with future databasing and **ontology** tools. These properties can be tracked using a relational graph or other database service, in a qualitatively similar manner to how transcripts are tracked across different versions of GENCODE genome builds (*Frankish et al., 2019*).

A major goal of the CCN is to track taxonomies and their associated cell sets by providing an easy-to-understand schema that is widely applicable to new and published taxonomies and that can be implemented through a user-friendly code base. The CCN is compatible with taxonomies generated from either single or multiple modalities, taxonomies applied to cells from overlapping datasets, and **reference taxonomies** (discussed in detail below). Each taxonomy is assigned a unique **taxonomy id** of the format CCN[YYYYMMDD][#], where 'CCN' denotes this nomenclature convention; Y, M, and D represent year, month, and day, respectively; and # is an index for compiling multiple taxonomies on a single day. Each taxonomy can also be assigned metadata, such as species, but such details are outside the scope of the CCN. Within each taxonomy, cell sets (and therefore also provisional cell types) are assigned multiple identifier tags, which are used for different purposes. **Cell set accession IDs** track unique cell sets across the entire universe of taxonomies and are defined as CS[YYYYMMDD][#]_[unique # within taxonomy], where CS stands for 'cell set' and the date and number match the taxonomy id. **Cell set labels** are useful for constructing cell sets from groups of provisional cell types, but can otherwise be ignored. **Cell set aliases** represent descriptors intended for public use and communication, including data-driven terms, historical names, or more generic cell type nomenclature. For convenience these are split into at most one **preferred alias**, which represents the primary tag for public consumption (e.g., the cell type names used in a manuscript), and

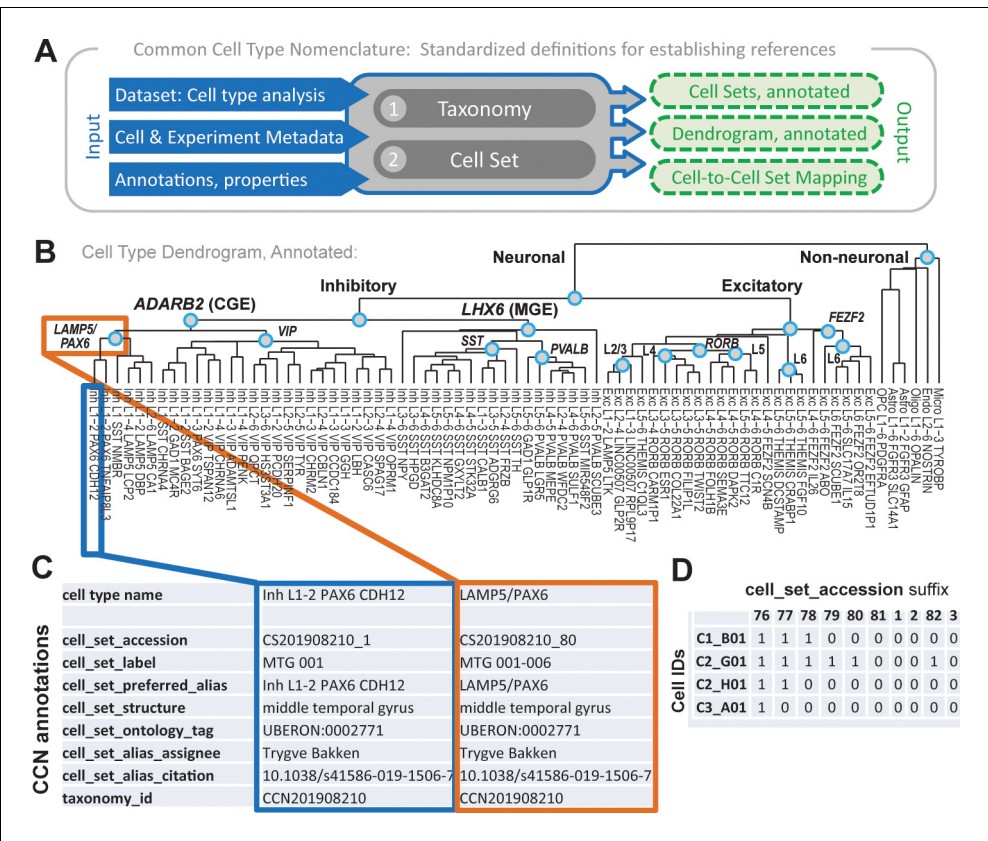

**Figure 1.** Overview of common cell type nomenclature (CCN) and application to human middle temporal gyrus (MTG). (**A**) Schematic of CCN components and process. (**B–D**) Example outputs from the CCN. (**B**) Annotated dendrogram of cell types in human MTG, along with associated cell type names, reproduced from *Hodge et al., 2019*. Internal nodes with a term (teal circles) represent cell sets with preferred alias tags. (**C**) CCN annotations for a putative cell type (outlined in blue) and an internal node (outlined in orange) of this dendrogram. (**D**) Snippet of an output file from the CCN showing cell to cell set mappings as applied to human MTG.

any other **additional aliases**. Additionally, each cell set can have at most one **aligned alias**, which is a biologically driven term that is selected from a controlled vocabulary. Aligned aliases generally are assigned to only a subset of cell sets by alignment to a reference taxonomy, but in principle can be assigned in any taxonomy or taxonomies (e.g., if a rare type is identified that is missing from the reference). The CCN includes a specific system for assigning such aliases in the mammalian cortex using properties that are predicted to be largely preserved across development, anatomical area, and species, which will be discussed in detail. Furthermore, the CCN includes a series of metadata tags tracking the provenance and anatomy of cell sets. The **cell set alias assignee** and **cell set alias citation** indicate the person and permanent data identifier associated with each cell set alias. The **cell set structure** indicates the location in the brain (or body) from where associated cells were primarily collected. Ideally, this will be paired to an established ontology using the **cell set ontology tag**; in this case, we use UBERON since it is designed as an integrated cross-species anatomy ontology (*Haendel et al., 2014*). Finally, the CCN is compatible with incorporation of additional taxonomy-specific or future global cell set metadata or descriptors. This could include donor metadata (e.g., age or sex), summarized cell metadata (e.g., cortical layer or average reads), or additional cell set tags. In particular, the concept of a cell set level is often useful for distinguishing highly specific but statistically less confident provisional cell types from the more general and more statistically robust cell sets.

The CCN is currently in use by the Allen Cell Types Database for transcriptomic taxonomies (http://celltypes.brain-map.org/rnaseq/) and is being applied to taxonomies generated by the BRAIN Initiative Cell Census Network (BICCN; https://biccn.org/) (*Bakken et al., 2020a*;

**Table 1.** Glossary of terms.

Terminology used with the common cell type nomenclature (CCN), definitions for use, and examples of how terms are applied. Terms are presented in bold upon first use in the text. This glossary is intended to clarify use for the purposes of the CCN since some terms are open to multiple interpretations, and effective classification requires disambiguation. Asterisks denote terms that represent specific components of the CCN.

| Term | Definition | Example |
|---|---|---|
| Taxonomy | Set of quantitatively derived data clusters defined by a specific computational algorithm on a specific dataset(s). Taxonomies are given a unique label and can be annotated with metadata about the taxonomy, including details of the algorithms and relevant cell and cell set IDs. | Any clustering result in a cell type classification manuscript |
| Dataset | Feature information (e.g., gene expression) and associated metadata from a set of cells collected as part of a single project. | Gene expression from 6000 human MOp nuclei |
| Ontology | A structured controlled vocabulary for cell types. | Cell Ontology |
| Marker gene(s) | A gene (gene set) which, when expressed in a cell, can be used to accurately assign that cell to a specific cell set. | GAD2; PVALB; CHODL |
| Taxonomy ID* | An identifier uniquely tagging a taxonomy of the format CCN[YYYYMMDD][#]. | CCN201910120 |
| Cell | A single entry in a taxonomy representing data from a single cell (or cell compartment, such as the nucleus). Cells have metadata including a unique ID. | N/A |
| Cell set | Any tagged group of cells in a taxonomy. This includes cell types, groups of cell types, and potentially other informative groupings (e.g., all cells from one donor, organ, cortical layer, or transgenic line). Cell sets have several IDs and descriptors (as discussed below) and can also have other metadata. | A cell type; a group of cell types; all cells from layer two in MTG; all cells from donor X |
| Provisional cell type | Quantitatively derived data cluster defined within a taxonomy. This is a specific example of a cell set that is of high importance, as most other cell sets are groupings of one or more provisional cell types. Here, the term 'cell type' is synonymous with 'provisional cell type.'. | A cell type defined in a specific study |
| Dendrogram | A hierarchical organization of provisional cell types defined for a specific taxonomy. Dendrograms have a specific semantic and visualizable structure and include nodes (representing multiple provisional cell types) and leaves (representing exactly one). Not all taxonomies include a dendrogram (e.g., if the structure of cell sets is non-hierarchical). | N/A |
| Community structure | Non-hierarchical relationships between cell types defined as groups of cell types in a graph. | N/A |
| Cell set accession ID* | A unique ID across all tracked datasets and taxonomies. This tag labels the taxonomy and numbers each cell type. CS[taxonomy id]_[unique # within taxonomy] | CS201910120_1 |
| Cell set label* | An ID unique within a single taxonomy that is used for assigning cells to cell sets defined as a combination of multiple 'provisional cell types'. | MTG 12 MTG 01–08 |
| Cell set alias* | Any cell set descriptor. It can be defined computationally from the data, or manually based on new experiments, prior knowledge, or a combination of both. Cell aliases beyond the 'preferred' or 'aligned' are defined as 'cell set additional aliases'. | (Any 'cell set aligned alias'); Interneuron 1; Rosehip |
| Cell set preferred alias* | The primary cell set alias (e.g., what cell types might be called in a publication). This can sometimes match the aligned alias, but not always, and can be left unassigned. | Inh L1-2 PAX6 CDH12; ADARB2 (CGE); Chandelier; [blank] |
| Cell set aligned alias* | Analogous to 'gene symbol'. At most one biologically driven name for linking matching cell sets across taxonomies and with a reference taxonomy. | L2/3 IT 4; Pvalb 3; Microglia 2 |
| Cell set structure* | The location in the brain (or body) from where cells in the associated set were primarily collected. | Neocortex |
| Cell set ontology tag* | A tag from a standard ontology (e.g., UBERON) corresponding to the listed cell set structure. | UBERON:0001950 |
| Cell set alias assignee* | Person responsible for assigning a specific cell set alias in a specific taxonomy (e.g., the person who built the taxonomy or uploaded the data, or a field expert). | (First author of manuscript) |
| Cell set alias citation* | The citation or permanent data identifier corresponding to the taxonomy where the cell set was originally reported. | (Manuscript DOI); [blank] |
| Reference taxonomy | A taxonomy based on one or a combination of high-confidence datasets, to be used as a baseline of comparison for datasets collected from the same organ system. | Cross-species cortical cell type classification |
| Morpho-electric(ME) type | A provisional cell type defined using a combination of morphological and electrophysiological features. | ME_Exc_7 |
| Governing body | A forum of subject-matter experts to guide policy and manage change of the CCN and associated ontologies and databasing efforts. | N/A |

*Adkins et al., 2020*; *Yao et al., 2020a*), a consortia of centers and laboratories working collaboratively to generate, analyze, and share data about brain cell types in human, mouse, macaque, and other non-human primates.

## Application of the CCN to cell types in human middle temporal gyrus

A detailed walk-through of how to apply the CCN to a published study on cell types in human middle temporal gyrus (MTG) (*Hodge et al., 2019*) is presented in Materials and methods. In short, *Figure 1B* recapitulates the cell types and associated hierarchy previously published for MTG (*Hodge et al., 2019*). After applying the CCN, each leaf (provisional cell type) and internal node of the dendrogram is assigned the series of cell set tags described above (*Figure 1C*), and every cell is mapped to every cell set (*Figure 1D*). This was all done using a user-friendly set of scripts (https://github.com/AllenInstitute/nomenclature). These output files are intended to be directly included as supplemental materials in manuscripts performing cell type classification in any species, and such output for human MTG (and for 17 additional taxonomies) is presented in *Supplementary file 1*.

## Naming cell types in mammalian cortex

Mammalian brain cell types inhabit a complex landscape with fuzzy boundaries and complicated correspondences between species and modalities, leading to a variety of disparate solutions for naming cell types. Thus, a challenging and potentially contentious question in cell type classification is how these newly identified cell types should be named, or in the context of the CCN, what should be put in the 'cell set aligned alias' identifier. The CCN utilizes a strategy for naming cell types in the mammalian cortex that includes properties that are cell intrinsic and potentially well conserved between species (*Table 2*). This convention is used as the cell set aligned alias tag in the CCN and ideally should directly map to cell types defined in a relevant ontology (i.e., Cell Ontology [*Diehl et al., 2016*] or Neuron Phenotype Ontology [*Gillespie et al., 2020*]). While admittedly underdeveloped, this convention has been applied to multiple studies of the primary motor cortex (M1; as discussed below) and represents only a starting point for discussion.

For glutamatergic neurons, cell types are named based on predominant layer(s) of localization of cell body (soma) and their predicted projection patterns. The relatively robust laminarity of glutamatergic cell types has been described based on cytoarchitecture in multiple mammalian species for many years (e.g., *Rakic, 1984*), and has been confirmed using RNA in situ hybridization (*Hodge et al., 2019*; *Tasic et al., 2018*; *Zeng et al., 2012*), and a combination of layer dissections and scRNA-seq (*Hodge et al., 2019*; *Tasic et al., 2018*). While in humans many cell types do not follow the layer boundaries defined by cytoarchitecture entirely, laminar patterning is still generally well conserved between human donors and mice (*Hodge et al., 2019*). In adult mouse visual cortex, projection targets for cell types have been explicitly measured using a combination of retrograde labeling and scRNA-seq (*Tasic et al., 2018*; *Tasic et al., 2016*). By aligning cell types across species, the projection targets in mice can be hypothetically extrapolated to putative projection targets in human or other mammalian species. For example, von Economo neurons are likely to project subcortically (*Hodge et al., 2020*). For GABAergic interneurons, developmental origin may define cell types by their canonical marker gene profile established early in development, with *Pvalb* and *Sst* labeling cell types derived from the medial ganglionic eminence and *Vip*, *Sncg*, and *Lamp5* labeling cell types derived from the caudal ganglionic eminence (*DeFelipe et al., 2013*). Non-neuronal cell types have not been a focus of the studies cited and hence they are labeled at a broad cell type level only. However, knowledge from other single-cell transcriptomics studies on microglia (*Hammond et al., 2019*; *Li et al., 2019*), astrocytes (*Batiuk et al., 2020*), and oligodendrocytes

**Table 2.** Proposed strategy for naming cortical cell types.

| Class | Format | Example |
| --- | --- | --- |
| Glutamatergic | [Layer] [Projection] # | L2/3 IT 4 |
| GABAergic | [Canonical gene(s)] # | Pvalb 3 |
| Non-neuronal | [Cell class] # | Microglia 2 |
| Any class | [Historical name] # | Chandelier 1 |

(*Marques et al., 2016*) could be included in subsequent versions of this convention. In all cases, multiple cell types are present within a given class. While it may not be possible to directly translate every feature to other brain structure or other organs, most of the concepts proposed here could still be followed.

## Alignment of established cell sets using reference taxonomies

The CCN presents a flexible data structure for storing key information about taxonomies and cell sets, implemented through reproducible code with standard output files, along with a specific convention for naming mammalian neocortical cell types. It is applicable to taxonomies defined on any data type using any classification algorithm, including hierarchical cell type classification using scRNA-seq. While useful for these reasons alone, a primary utility of the CCN is to facilitate cross-study integration of cell type classifications, in particular when applied in the framework of a reference taxonomy. A reference taxonomy (or reference cell type classification) is any taxonomy based on one or a combination of high-confidence datasets, which can be used as a baseline of comparison for other datasets collected from the same organ system. For example, many researchers favor building a gene expression-based reference taxonomy based on high-throughput, high-resolution single-cell transcriptomics assays and then layering on additional phenotypic data as they become available (*Yuste et al., 2020*). Molecular, physiological, and morphological characteristics of cortical neurons are highly correlated based on simultaneous measurement in individual cells using Patchseq (*Berg et al., 2020*; *Gouwens et al., 2020*; *Scala et al., 2020*), making such a strategy feasible. Many groups are currently performing scRNA-seq analysis in different areas of the brain, from all organs in the human body (*Rozenblatt-Rosen et al., 2017*), from multiple mammalian species (*Geirsdottir et al., 2019*), and across trajectories of development (*Nowakowski et al., 2017*), aging (*Tabula Muris Consortium, 2020*), and disease (*Mathys et al., 2019*). Application of the CCN to these datasets will allow future reference taxonomies to evolve to accommodate these additional complexities by overlaying a common data structure and associated nomenclature.

Reference taxonomies and the CCN are two components of a multi-staged analysis workflow for aligning cell type classifications using datasets collected across multiple labs, from multiple experimental platforms, and from multiple data modalities (*Figure 2*). This workflow accommodates methodological differences in cell type definitions across studies and accommodates changes in reference taxonomies over time. The proposed workflow can be broken down into four broad stages:

1. First, many research teams will independently define cell types, identify their discriminating features, and name them using one of many available experimental and computational strategies. This represents the current state of the field. The CCN may be applied to each dataset independently at this stage.
2. Second, an initial reference cell type classification will be defined by taking the results from one or more (ideally validated) datasets and integrating these data together in a single analysis, if needed. Being high dimensional, high throughput, and relatively low cost, transcriptomics strategies are immediately applicable to many organs and species, and the goal is for reference cell types to be defined using this modality (*Yuste et al., 2020*). The CCN will then be applied to the reference taxonomy as described above – the CCN treats reference taxonomies identically to any other taxonomy. Importantly, aligned aliases should be defined in the reference taxonomy at this stage using a standard naming convention such as the one proposed above.
3. This reference cell type classification can now be used as a comparator for any related datasets, providing a mechanism for transferring prior knowledge about cell types across datasets. Cell sets from existing taxonomies can be renamed using one of the many validated alignment algorithms (e.g., *Barkas et al., 2018*; *Butler et al., 2018*; *Gala et al., 2019*; *Johansen and Quon, 2019*) by integrating data from this taxonomy with the reference, and then updating the cell set aligned alias to match terms defined in the reference. For new datasets, taxonomies can be generated using any clustering or alignment strategy followed by the same mapping and annotation transfer steps.
4. Finally, new versions of the reference cell type classification should be periodically generated using additional data and/or computational methods, and this new classification will now be used as comparator for related datasets. Steps 3 and 4 can iterate at some to-be-defined cadence.

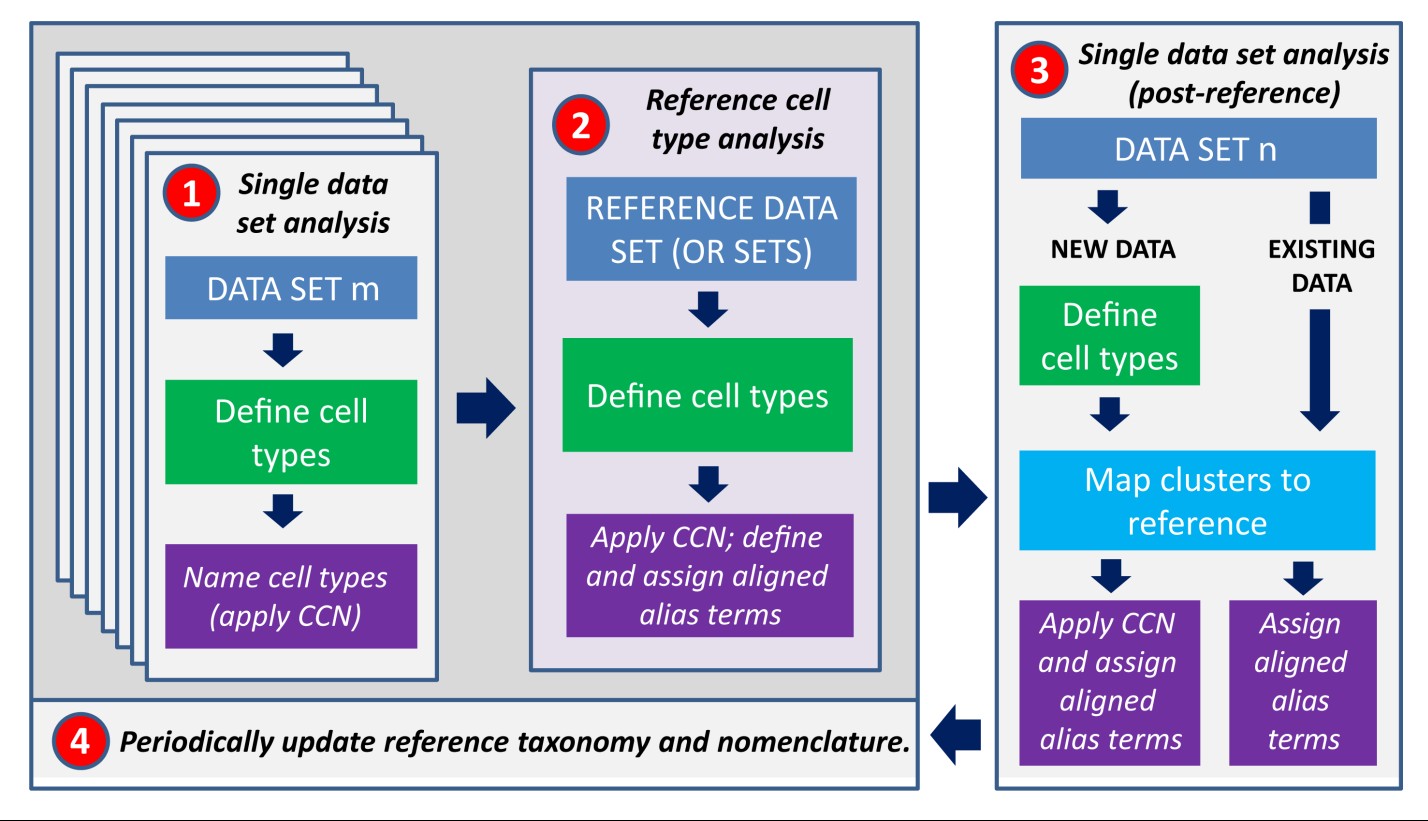

**Figure 2.** Workflow for assigning types to a given dataset with taxonomy. (1) Cell type classification will initially be performed separately on all taxonomies. (2) One, some, or all of these datasets will be combined into a high-confidence reference taxonomy which can be used as a comparator for any related datasets, by (3) mapping existing and new datasets to the reference taxonomy. (4) The reference will periodically be updated as new datasets and taxonomies are generated.

This workflow provides two complementary strategies to compare between taxonomies without needing to look at gene expression or other quantitative features. First, each taxonomy draws upon a common set of aligned alias terms, which allows for immediate linking of common cell sets between taxonomies (in cases where such information can be reliably assigned). A second strategy is through inclusion of common datasets across multiple taxonomies (reference or otherwise); if cells are assigned to the same cell sets in more than one taxonomy, then the cell sets can be directly linked. As a whole, this workflow provides a general outline for versioned cell type classification that could be specialized as needed for communities studying different organ systems and that provides a starting point for design of future cell taxonomy and nomenclature databases.

## Defining a cross-species reference taxonomy in M1

A recent study profiling nearly half a million nuclei in primate and mouse primary motor cortex (M1) presents a taxonomy suitable for defining as a reference taxonomy (*Bakken et al., 2020a*). This study included single cell data from three separate 'omics' modalities (transcriptomics, epigenetics, and methylation) for mouse, marmoset, and human. Datasets were integrated in two ways. First, epigenetics and methylation datasets were integrated with snRNA-seq data within mouse, marmoset, and human independently (as shown in *Figure 3A* for human), which demonstrates a consistent genomic profile of cell types within species. Second, snRNA-seq from each species were aligned into a single integrated reference, which identifies cell type homologies across species that were presumably present in the mammalian ancestor to rodents and primates. This evidence-based assumption of cross-species homologies provides a strategy for transferring cell type characteristics from rodent studies (e.g., projection targets) into human, where experiments for making such

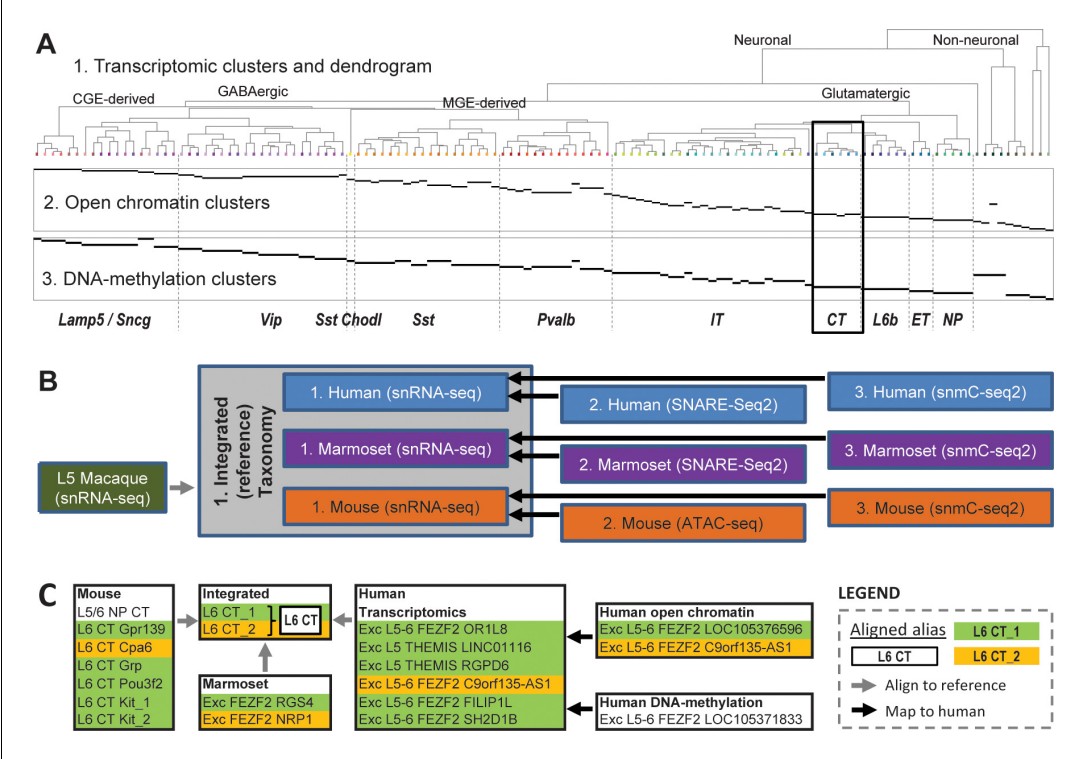

**Figure 3.** Series of multimodal, cross-species taxonomies in primary motor cortex (M1) demonstrates utility of nomenclature schema. (**A**) Taxonomies based on transcriptomic ('1'; top), open chromatin ('2'; middle), and DNA methylation ('3'; bottom) in human M1. Epigenomic clusters ('2', '3'; in rows) aligned to RNAseq clusters ('1') as indicated by horizontal black bars and are also assigned matching cell sets in the relevant taxonomies. Adapted from *Bakken et al., 2020a*. (**B**) Flow chart showing all 11 taxonomies generated for this project and their connections. The integrated (reference) taxonomy included nuclei collected using snRNA-seq from three species (gray box), with nuclei collected from layer five in macaque mapped to this space post hoc (gray line). Separately, epigenetics taxonomies from human, marmoset, and mouse were aligned to their respective transcriptomics taxonomies (black lines). This entire taxonomic structure is captured by the CCN (see *Supplementary file 1*). (**C**) An example mapping of corticothalamic (L6 CT) provisional cell types across the human and transcriptomics taxonomies using the CCN (black box in **A**). Preferred aliases for each taxonomy are used for clarity.

measurements are not yet possible. A total of 11 taxonomies were generated (*Figure 3B*), and all were included in the same nomenclature schema, and the CCN was applied to this set of taxonomies as described above (see Supplementary Table 3 in *Bakken et al., 2020a* and *Supplementary file 1*). *Figure 3C* shows an example of these cross-species and cross-modality alignments for L6 CT cells, which are divided into two cell sets in the integrated taxonomy (and assigned the aligned alias tags L6 CT_1 and L6 CT_2) and includebetween one and seven cell sets in the single-modality taxonomies.

This integrated taxonomy (*Figure 3B*, gray box) represents a suitable reference taxonomy for several reasons: first, the data generation, data analysis, and write-up spanned multiple BICCN-funded labs as part of a coordinated consortium project, indicating that this taxonomy was approved by a large subset of the neocortex cell typing community; second, while a number of differences were found between species, 45 core provisional cell types could be aligned across all species with transcriptomics; third, the taxonomies generated using epigenetics and methylation are largely consistent with results of this integrated taxonomy (*Figure 3A*, bottom panels and *Yao et al., 2020a*); and finally, this taxonomy can be linked with other quantitative features (such as morphology, electrophysiology, and expected projection targets) through comparison with mouse studies using complementary modalities such as Patch-seq (*Gouwens et al., 2020*; *Scala et al., 2020*) and Retro-seq (*Tasic et al., 2018*; *Tasic et al., 2016*). Using these linkages, aligned aliases of the format proposed in *Table 2* were assigned to cell sets in the integrated taxonomy along with the 10 other species-specific taxonomies using a combination of (1) robust gene markers from the literature, (2) highly

discriminating gene markers in these data, (3) projection targets in mouse, (4) historical names based on cell shape, and (5) broad or low-resolution cell type names (that directly map to ontologies), providing a starting point for how brain cell types could be named. A complete list of aligned aliases used is shown in *Supplementary file 2*.

## Applying the CCN to existing and new datasets

For a specific convention to be adopted, both in general or in place of other competing conventions, it needs to be easy to use and immediately useful to the community. For example, many cell type classification studies now use Seurat (*Butler et al., 2018*) for clustering and alignment because it produces believable biological results, and it is implemented in intuitive R code with extensive user guides for non-specialists. As such, Seurat visualizations appear frequently in manuscripts and its file format is used as input for several analysis pipelines. While the usability of the CCN has been established above, the utility of applying it to a single taxonomy in the absence of a centralized database of taxonomies may be less clear. Here five use cases are presented to illustrate how the CCN can be applied to published datasets using scRNA-seq and electrophysiology and morphology in multiple species. These use cases provide immediate utility and also lay a foundation for future databasing and ontology efforts.

### Use case 1: Alignment of human MTG taxonomy to M1 reference

The M1 reference taxonomy includes a validated set of aligned aliases that follows the proposed nomenclature for mammalian cortex (*Table 2*) and that can be applied to any other taxonomy. As part of the original analysis (*Bakken et al., 2020a*), nuclei from human MTG (*Hodge et al., 2019*) were aligned to the human M1 dataset. This analysis provides a perfect use case for transferring cell set aligned alias tags from the reference taxonomy to the MTG data (as was done; see Materials and methods). *Figure 4* shows a visualization of glutamatergic types in M1 and MTG, with the color of each square representing the fraction of cells from provisional cell types in each brain region that are assigned to the same alignment cluster, and boxes indicating the aligned alias calls in M1 and their corresponding calls in MTG. While alignment is not perfect for provisional cell types, it is sufficient for matching aligned aliases between cortical areas. These mappings enable biological insights such as the presence of L4-like neurons in M1, where an anatomically defined L4 is not apparent. Likewise, such alignment enables prediction of cell properties such as long-range connectivity (e.g., thalamic inputs), as well as electrophysiology measurements in primary sensorimotor cortices or other brain regions inaccessible to techniques such as Patch-seq. Similar alignments were performed for GABAergic interneurons and non-neuronal cell types (*Supplementary file 1*). Such tagging allows cell sets in human MTG to be directly compared to cell sets from any other taxonomy with the same aligned alias, for example to infer morphological or electrophysiological properties (see Use case 2) or cell class persistence across multimodal phenotypes and developmental stages (see Use case 3) in mouse. Cell sets can even be matched with more distant species using the CCN (see Use case 4), to the extent that such alignment is possible based on the data.

### Use case 2: Building a morpho-electric taxonomy

While much effort for cell typing is currently focused on taxonomies based on scRNA-seq datasets, the CCN can equally apply to non-transcriptomic and non-hierarchical taxonomies. For example, a study of mouse visual cortex examined ~1800 cells characterized electrophysiologically by whole-cell patch clamp recordings, and for a subset of these (450 cells), morphological reconstructions were also performed (*Gouwens et al., 2019*). Using a multimodal unsupervised clustering method, the authors identified 20 excitatory and 26 inhibitory **morpho-electric types** (or **me-types**), which are cell types defined using a combination of morphological and electrophysiological features. *Figure 5* shows the application of the CCN to a subset of excitatory (glutamatergic) me-types of that study (see *Supplementary file 1* for application to remaining me-types). The preferred alias and inferred subclass columns show the organization scheme; me-types were organized by broader cell types inferred from transgenic labels, but not placed into a binary hierarchical taxonomic tree (*Gouwens et al., 2019*). Through application of the aligned alias tag, these cell types can be directly linked to cell types defined based on transcriptomics.

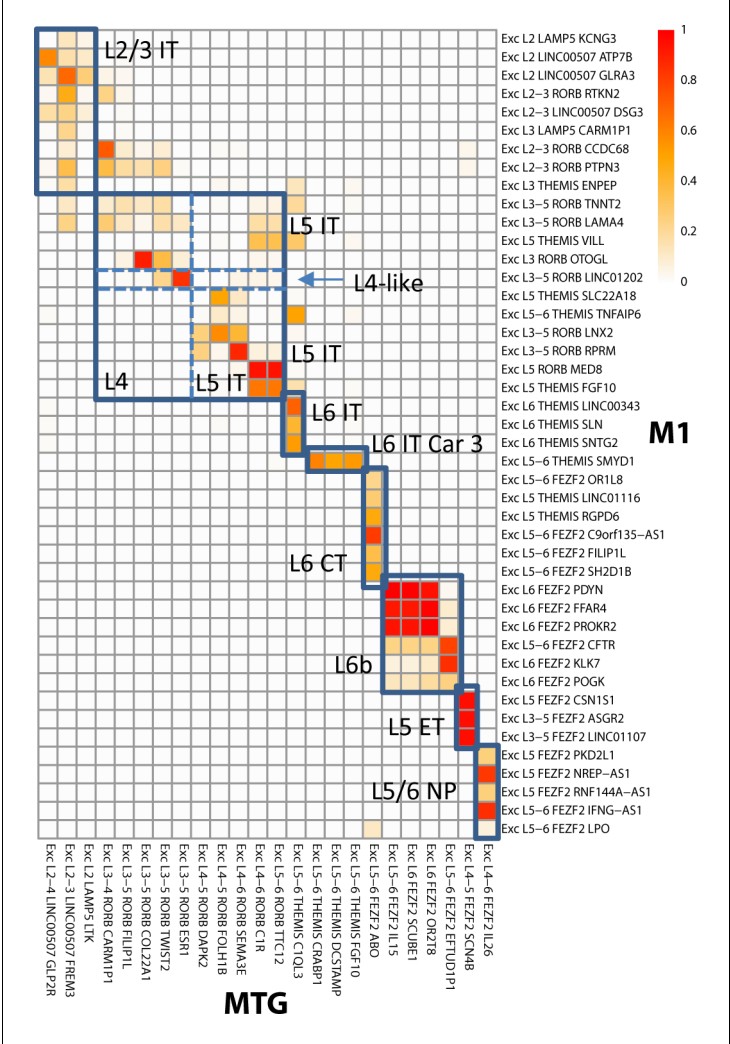

**Figure 4.** Alignment of glutamatergic cell sets in human middle temporal gyrus (MTG) to a reference primary motor cortex (M1) taxonomy. Cluster overlap heatmap showing the proportion of nuclei from MTG clusters and the reference (M1) clusters that coalesce with a given aligned cluster. Cell sets corresponding to aligned aliases in the MTG and M1 taxonomies are labeled and indicated by blue boxes. Adapted from *Bakken et al., 2020a*.

## Use case 3: Exploring an interneuron subclass using multimodal attributes: The 'Sst Chodl' class persists across cross-taxonomy matching

Somatostatin-expressing interneurons can be divided into multiple cell types (the specific number differs by taxonomy), some of which include cells that express Chodl in the mouse cerebral cortex (*Tasic et al., 2018*; *Tasic et al., 2016*). These 'Sst Chodl' neurons are rare and, based on expression of specific marker genes, correspond to the only known cortical interneurons with long-range projections (*Tomioka et al., 2005*). Recent studies using the multimodal cell phenotyping method Patch-seq (*Gouwens et al., 2020*) confirmed that 'Sst Chodl' cell sets characterized based on morphology and electrophysiology (*Gouwens et al., 2019*) match those defined by transcriptomic profiles (*Tasic et al., 2018*; *Tasic et al., 2016*). The CCN can be applied to readily represent these 'Sst Chodl' cells (and other cell types) matched between all relevant taxonomies, regardless of species or modality through the use of aligned alias tags. For example, *Table 3* shows all cell sets from *Bakken et al., 2020a* (*Figure 3B*) associated with Sst Chodl cells, which all have 'Sst Chodl' in the aligned alias (with one exception noted below). In mouse, all three modalities have a single 'Sst Chodl' cell type, which can be linked to a matched type in VISp due to its highly distinct gene expression patterning that is conserved across brain regions (*Tasic et al., 2018*; *Yao et al., 2020b*).

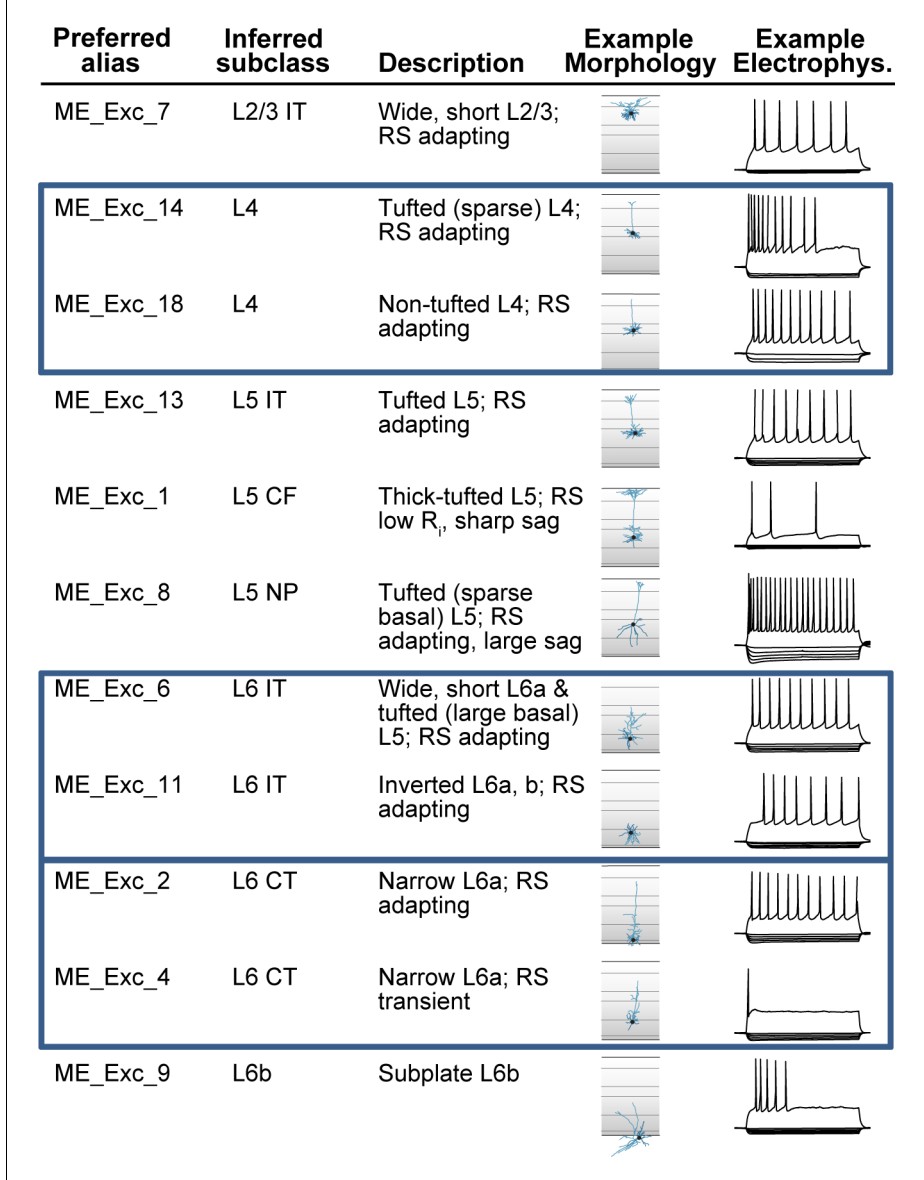

**Figure 5.** Application of common cell type nomenclature (CCN) to glutamatergic me-types in the mouse visual cortex. Excitatory (glutamatergic) me-types from *Gouwens et al., 2019* that have been incorporated into the nomenclature schema. Eleven of the original 20 excitatory me-types are shown as examples. Representative morphologies and electrophysiological responses are shown to illustrate the differences between types. The 'inferred subclass' calls perfectly map to cell set aligned aliases from the reference M1 taxonomy in *Figure 3*, except that L5 CF (corticofugal) is an additional alias for L5 ET, and cells sets corresponding to L4, L6 IT, and L6 CT (blue boxes) have been added to the taxonomy.

This transcriptomic cell type is similarly linked to the 'Sst Chodl' cell type in the integrated transcriptomic (reference) taxonomy, which lists 'long-range projecting Sst' as an additional alias to formalize the cross-modal correspondence. In human, the RNA-seq and ATAC-seq have one-to-one correspondences, but for DNA methylation (DNAm), Inh L1-5 SST AHR aligns with several Sst cell types including Sst Chodl (likely due to the rarity of this cell type). Cell sets from the methylation- and epigenetics-based taxonomies include an additional alias that list the cell set labels in transcriptomics taxonomy, directly linking these cell types. Therefore, while Inh L1-5 SST AHR does not have 'Sst Chodl' as its aligned alias, the cell set label 'RNA-seq 040, 046–047, 050–052, 068 in CCN201912131' indicates the inclusion of 'Sst Chodl' cells (RNA-seq 040). In marmoset, where fewer

**Table 3.** Nomenclature for 'Sst Chodl' cell sets cited in **Bakken et al., 2020a**.

Relevant common cell type nomenclature (CCN) entities and taxonomy metadata, including the cell set additional alias that links to cell set labels from relevant transcriptomics taxonomies. All listed cell sets have a cell set structure of 'primary motor cortex' and a cell set ontology tag of 'UBERON:0001384'.

| # | Cell set preferred alias | Cell set label | Cell set accession | Cell set aligned alias | Cell set additional alias |
|---|---|---|---|---|---|
| 1 | Inh L1-6 SST NPY | RNA-seq 040 | CS201912131_40 | Sst Chodl | |
| 2 | Inh L1-5 SST AHR | DNAm 12 | CS202002272_12 | | RNA-seq 040, 046–047, 050–052, 068 in CCN201912131 |
| 3 | Inh L1-6 SST NPY | ATAC-seq 08 | CS202002273_8 | Sst Chodl | RNA-seq 040 in CCN201912131 |
| 4 | Inh SST NPY | RNA-seq 01 | CS201912132_1 | Sst Chodl | |
| 5 | Sst Chodl | RNA-seq 028 | CS202002013_28 | Sst Chodl | |
| 6 | Sst Chodl | DNAm 09 | CS202002276_9 | Sst Chodl | RNA-seq 028 in CCN202002013 |
| 7 | Sst Chodl | ATAC-seq 10 | CS202002277_10 | Sst Chodl | RNA-seq 028 in CCN202002013 |
| 8 | Sst Chodl | Integrated 14 | CS202002270_14 | Sst Chodl | Long-range projecting Sst |

| # | Cell set alias assignee | Cell set alias citation | Taxonomy id | Species | Modality |
|---|---|---|---|---|---|
| 1 | Nikolas Jorstad | 10.1101/2020.03.31.016972 | CCN201912131 | Human | RNA-seq |
| 2 | Wei Tian | 10.1101/2020.03.31.016972 | CCN202002272 | Human | DNAm |
| 3 | Blue Lake | 10.1101/2020.03.31.016972 | CCN202002273 | Human | ATAC-seq |
| 4 | Fenna Krienen | 10.1101/2020.03.31.016972 | CCN201912132 | Marmoset | RNA-seq |
| 5 | Zizhen Yao | 10.1101/2020.02.29.970558 | CCN202002013 | Mouse | RNA-seq |
| 6 | Hanqing Liu | 10.1101/2020.02.29.970558 | CCN202002276 | Mouse | DNAm |
| 7 | Yang Li | 10.1101/2020.02.29.970558 | CCN202002277 | Mouse | ATAC-seq |
| 8 | Nikolas Jorstad | 10.1101/2020.03.31.016972 | CCN202002270 | All | RNA-seq |

cells were collected, an 'Sst Chodl' cell set is only found with transcriptomics. Explicitly linking cell sets in this way provides multiple potential points of comparison with other studies, including studies of disease or development. For example, a study of interneuron development in E14 mice found that the 'Sst Chodl' cells were severely affected by Sox6 removal during interneuron migration (**Munguba et al., 2019**), and cell class definitions observed in the mature brain may have foundational roles in cortical patterning.

## Use case 4: Alignment of cell types from reptilian and mammalian cortex using the CCN

While the focus of this study is the mammalian cortex, the CCN framework is applicable to other organs and more distant species. As an example use case, a single-cell transcriptomics study of turtle and lizard pallium found GABAergic interneuron and non-neuronal cell types to be homologous with those seen in mouse cortex (**Tosches et al., 2018**). In many cases, these cell types expressed shared gene markers, suggesting a shared evolutionary origin across 320 million years of evolution in amniote vertebrates. These types include astrocytes (GFAP), oligodendrocytes (MBP), oligodendrocyte precursor cells (OLIG1 and PDGFRA), microglia (C1QC), GABAergic interneurons as a whole (GAD1 and GAD2), and Sst+ interneurons (SST). Reptilian analogs for other CGE- and MGE-derived GABAergic types were also identified, although interestingly neither VIP nor PVALB marker genes are expressed in reptiles. Application of the CCN to the taxonomies presented for the turtle demonstrates the utility of this approach (**Supplementary file 1**).

Assignment of aligned aliases for non-neuronal cells and GABAergic interneurons is straightforward, with 'PV-like' interneurons (cell types i11–i13 from **Tosches et al., 2018**) assigned 'PVALB', and similar alignments for other types. In contrast, the correspondence between reptilian and mammalian glutamatergic cells is more complicated. Reptiles have a three layer pallium and only the anterior dorsal cortex (representing a small fraction of pallium) is comparable with the six-layer mammalian neocortex (**Jarvis, 2009**; **Tosches et al., 2018**). RNA-seq in combination with in situ hybridization identified two distinct sublayers of turtle layer 2: a superficial L2a (cell types e07–e08) and a

deeper L2b (e13–e16), which seem to correspond with mammalian deep layer and upper layer neurons, respectively, suggesting there was likely an inversion of layers in one clade. However, all of these cell types coexpress genes found in mutually exclusive L2/3, L4, and L5a intra-telencephalic neurons (e.g., SATB2, RORB, and RFX3) along with extra-telencephalic projection neurons (e.g., BCL11B, TBR1, and SOX5), suggesting either a lack of homologous cell types between clades or at least a change in the core transcription factor regulatory programs. Thus, with the level of resolution presented in this study, no aligned aliases (beyond the broadest) are assigned for glutamatergic types. This points to the importance of having measurements in other modalities, for example, local and long-range connectivity, that may help establish homologies or bolster claims of clade-specific cellular innovations. Overall, the CCN provides a mechanism for assigning a standard nomenclature for cell types found in the reptilian cortex and linking these types with a mammalian neocortical reference at the level of resolution resolved in the taxonomy.

## Use case 5: Comparison of novel to existing taxonomies

The first four use cases represent specific examples of how taxonomies from different brain regions, modalities, and species can be presented in the framework of the CCN to make published inferences more easily accessible to a naive reader. These represent specific examples of a more general use case for scientists, who may want to compare their newly generated taxonomy to what is currently known about cell types. The ideal application for this scenario is a centralized database for taxonomy integration with an associated ontology and annotation capabilities; such a framework is well beyond the scope of this manuscript, but solutions are underway. As a starting point for this goal, *Supplementary file 1* presents output files from the CCN for 18 taxonomies (including all taxonomies discussed herein; *Table 4*) that have been annotated with the aligned aliases from the M1 reference taxonomy presented in *Figure 3*. Transcriptomics-based taxonomies were collected from human, non-human primate, mouse, and reptile, and span multiple neocortical areas. In addition, several of these are matched to taxonomies collected using other modalities such as morphology,

**Table 4.** Taxonomies with applied CCN.

Table showing the set of taxonomies included in *Supplementary file 1*. All taxonomies include the annotated nomenclature table. Asterisk (*) and carrot () indicate that the updated dendrogram and cell to cell set mapping files are also included for that taxonomy, respectively. CCN202002270 is the reference taxonomy presented in *Figure 3B*.

| Taxonomy id | Description | Reference |
|---|---|---|
| CCN201810310* | Mouse VISp + ALM (from the *Tasic et al., 2018*) | *Tasic et al., 2018* |
| CCN201908210* | Human MTG (from the *Tasic et al., 2018*) | *Hodge et al., 2019* |
| CCN201908211* | Joint mouse/human analysis (slight modification from *Hodge et al., 2019*) | *Hodge et al., 2019* |
| CCN201912130 | Human M1 taxonomy using 10× data | *Bakken et al., 2020a* |
| CCN201912131 | Human M1 taxonomy using Smart-seq and 10x data | *Bakken et al., 2020a* |
| CCN201912132 | Marmoset M1 taxonomy using 10× data | *Bakken et al., 2020a* |
| CCN202002013* | Mouse MOp BICCN taxonomy using multiple RNAseq datasets | *Yao et al., 2020a* |
| CCN202002270 | Cross species (integrated) transcriptomics taxonomy | *Bakken et al., 2020a* |
| CCN202002271 | Macaque transcriptomics taxonomy, layer 5/6 only | *Bakken et al., 2020a* |
| CCN202002272 | Human DNA methylation taxonomy | *Bakken et al., 2020a* |
| CCN202002273 | Human ATAC-seq taxonomy | *Bakken et al., 2020a* |
| CCN202002274 | Marmoset DNA methylation taxonomy | *Bakken et al., 2020a* |
| CCN202002275 | Marmoset ATAC-seq taxonomy | *Bakken et al., 2020a* |
| CCN202002276 | Mouse DNA methylation taxonomy | *Yao et al., 2020a* |
| CCN202002277 | Mouse ATAC-seq taxonomy | *Yao et al., 2020a* |
| CCN202005150 | Mouse inhibitory neurons in VISp defined using electrophysiology, morphology, and transcriptomics | *Gouwens et al., 2020* |
| CCN201906170 | Mouse neurons in VISp defined using electrophysiology and morphology | *Gouwens et al., 2019* |
| CCN201805250 | Turtle pallium transcriptomics taxonomy | *Tosches et al., 2018* |

electrophysiology, epigenetics, and methylation. Such breadth provides multiple avenues of entry into this framework for annotation of novel datasets and allows for a more flexible implementation of the specific analysis workflow described in *Figure 2*. In particular, instead of requiring alignment of new datasets to the reference taxonomy, new datasets can be aligned with any taxonomy from *Table 4*, and information about cell type can then be inferred from any cell sets in any included taxonomy with a common aligned alias as the matched cell set. If this process is applied to novel taxonomies and the output files are included as supplemental materials in any resulting manuscript, then these taxonomies can be included in any future centralized database with minimal effort, providing a richer reference for further study.

## Discussion

The complexity of cell types taxonomies and their generation now requires conventions and methodology to capture and communicate essential knowledge derived from experiments. The CCN provides a schema and workflow that allows scientists to organize their cell types within a single dataset and to link taxonomies using the aligned alias and other alias terms. However, the CCN is currently a stand-alone nomenclature schema that lacks the centralization and governance of gene-based standards proposed by the HUGO Gene Nomenclature Committee (HGNC) (*Bruford et al., 2020*) and does not yet have a mechanism for integrating with underlying data and metadata.

These shortcomings would be addressed through linking cell type ontology curation with corresponding databases. Ontology curation would allow users to associate data-derived cell sets to common usage terms from prior knowledge, and connect directly with the well-annotated ontology tools that are available for broader classifications (e.g., the Cell Ontology, http://www.obofoundry.org/ontology/). In addition, aligned aliases defined in reference taxonomies could represent a starting point for expansion of existing ontologies to presumptive cell types defined using other data-driven approaches (such as the terms in *Table 2* for cortical neurons). Centralizing a location for taxonomies, their associated cell sets, and underlying datasets could provide a more robust ecosystem for comparing relevant nomenclature information, other metadata, and the primary data itself. Such databases can be implemented using knowledge graph-based models (*Alshahrani et al., 2017*; *Waagmeester et al., 2020*), permitting traversal across a *data, information, knowledge, and wisdom* hierarchy (*Rowley, 2007*). A potential presentation could be a 'Cell Type Card', instantiated as a web-accessible reference that compiles information about a specific cell set in a standardized summary. Not unlike a periodic table in structure, this concept has been implemented for genes (http://www.genecards.org), and as a prototype using transcriptomically defined cell types in mouse hippocampus and cortex (*Yao et al., 2020b*).

Incorporating community input on the definition and management of cell type standards will be necessary as new experiments are performed and additional evidence emerges. A cell type standards **governing body** would ideally be responsible for vetting ontologies for organizing data, controlled vocabulary for assigning cell type nomenclature, and will need to define a process for submission to ensure that critical data and metadata can be stored in a robust database. Deciding which taxonomies to include as reference taxonomies, along with frequency of updates, and how to address the breadth of brain regions, data modalities, cross-species reconciliation, and stochasticity of developmental and disease trajectories is essential. Organizing such a governance framework represents an important step and efforts are under way through BRAIN Initiative-funded initiatives, but is beyond the scope of work presented here.

This work presents a framework that is a modest step in a long and iterative process. With cross-disciplinary partnership and ever-increasing data, refinement of this proposed convention is expected. Together with collaborators, the Allen Institute has begun to combine ontology development, data integration, and nomenclature formalization efforts with the aim of facilitating cell type standards for the neuroscience community. Together with the goals articulated as part of the NIH BICCN and Brain Cell Data Center (BCDC) (https://biccn.org/), we seek to provide access to the diverse cell types in the human, mouse, and marmoset brain. The Allen Brain Map Community Forum (https://community.brain-map.org/c/cell-taxonomies/) has a dedicated space for discussion related to cell taxonomy refinement, to promote open and accessible opportunity for exchanging ideas and suggesting improvements. Beyond brain, whole-body projects seeking to categorize cell types, such as the NIH Common Fund-supported Human BioMolecular Atlas Program (HuBMAP, https://

hubmapconsortium.org/) and the Human Cell Atlas consortium (https://www.humancellatlas.org/), will also need to leverage organizational conventions such as this, for comparable purposes that are practical and promote scientific rigor. The authors look forward to engagement with emerging communities and forums as evolution of cell classification methods continues.

## Materials and methods

User-friendly executable code for applying the CCN is available on GitHub (https://github.com/AllenInstitute/nomenclature). This repository aims at providing a set of standardized terms and files that are immediately useful and also formatted to seed any future or in-process platform for cell type characterization and annotation. It is written as a user-friendly script in the R programming language (https://www.R-project.org) that includes specific details for how to apply the CCN, along with a set of example input files from a published study on cell types in human MTG (*Hodge et al., 2019*).

### Step-by-step application of the CCN to human MTG

This section addresses how to apply the CCN to an example taxonomy, from human MTG. Three inputs are required to run the CCN: (1) a cell type taxonomy (not necessarily hierarchical), (2) a cell metadata file with cluster assignments (and optionally additional information), and (3) optional manual annotations of cell sets (e.g., aliases), which typically would be completed during taxonomy generation. Example files for human MTG are saved in the repository's data folder. Once all files are downloaded and the workspace is set up, several global variables are set, which propagate to each cell set as a starting point, and which can be updated for specific cell sets later in the process. A unique taxonomy_id of the format CCN[YYYYMMDD][#] is chosen, which will match the prefix for cell set accession IDs. To ensure uniqueness across all taxonomies, taxonomy_ids are tracked in a public-facing database, with future plans to transfer these to a more permanent solution that will also provide storage for accompanying CCN output files and relevant metadata. In addition, values for the cell set assignee, citation, structure, and ontology tag are defined, along with the prefix(es) for the cell set label. For human MTG, 'CCN201908210', 'Trygve Bakken', '10.1038/s41586-019-1506-7', 'middle temporal gyrus', 'UBERON:0002771', and 'MTG' are used, respectively. Next, the dendrogram is read in as the starting point for defining cell sets by including both provisional cell types (terminal leaves) and groups of cell types with similar expression patterns (internal nodes). *Figure 1B* shows the annotated dendrogram in human MTG provided in the GitHub repository, under which are displayed the names of cell types presented in *Hodge et al., 2019*. These provisional cell types were named using an entirely data-driven strategy: (cell class) (L)(cortical layers of localization) (canonical marker gene) ([optional] specific marker gene), as discussed in *Hodge et al., 2019*.

The main script takes the preset values and dendrogram as input, assigns accession ids and labels for each cell set, and then outputs an intermediate table and a dendrogram with all CCN labels defined in the previous section (*Figure 1C*). By default, the provisional cell types are assigned their original name from the dendrogram as preferred alias (e.g., 'Inh L1-2 *PAX6 CDH12*'), while this field is left blank for internal nodes. For all cell sets, fields for additional and aligned alias are also initially left blank. Cell set labels are formatted as the label prefix (e.g., 'MTG') followed by a list of the cell set labels of all included provisional cell types. For example, the '*LAMP5/PAX6*' node in human MTG includes the first six cell types in the tree and therefore has the cell set label of 'MTG 001–006'. The table with these CCN tags for each cell set is then written to a csv file for manual annotation, which includes two critical aspects: (1) creation of new cell sets and (2) updating CCN tags for any cell sets. Cell sets corresponding to groups of relevant cell types either based on biological relevance (e.g., *LAMP5*-associated cell types in MTG) or as defined using a non-hierarchical computational strategy can be added at this step. In addition, cell sets corresponding to metadata rather than cell types can also be added. For example, in human MTG, 'CS201908210_154' corresponds to the set of nuclei collected from neurosurgical tissue and is given a cell set label of 'Metadata 1' and a preferred alias of 'Neurosurgical'.

After finalizing these cell sets, they can then be annotated to include additional aliases based on known literature (e.g., assigning 'basket' or 'fast-spiking' to relevant *PVALB+* cell sets), along with the assignees and citations from which such aliases were derived (e.g., 'Nathan Gouwens' and '10.1101/2020.02.03.932244'). As another example, Inh L1-4 *LAMP5 LCP2* corresponds to Rosehip

cells (see *Boldog et al., 2018*) and therefore an additional alias for this cell type is 'Rosehip'. The structures and associated ontology tags could also be updated at this stage. For example, previous studies in mice suggest that most non-neuronal and GABAergic cell types are conserved across cortical areas (*Tasic et al., 2018*; *Yao et al., 2020b*). Although not done here, relevant cell sets could be generalized to an anatomic structure such as 'Neocortex' (UBERON:0001950). A final component of manual annotation is to update relevant cell sets with an aligned alias (e.g., a common usage term), which is critical for comparison of taxonomies in the CCN. In this case, aligned aliases for all cell sets were assigned by comparison with the human M1 reference (*Bakken et al., 2020a*), as shown in *Figure 4* for glutamatergic neurons and as described in Use case 1. It is important to note that this step requires a previous computational alignment (or some other strategy to match cell sets) to use as evidence prior to assignment of the aligned alias; cell set alignment itself is not performed as part of the CCN.

After completing the manual annotations, the updated table is read back into R for additional dendrogram annotation and for mapping of cells to cell sets. Dendrograms are revised to include the new cell sets and annotations, and then output in a few standard formats (.RData, .json, and .pdf) for ontology construction and other downstream uses. Individual cells are then mapped to cell sets using the cell metadata table, which includes a unique cell identifier, provisional cell type classification, and other optional metadata. Cells are then mapped to cell sets representing one or more provisional cell type using the annotated dendrogram and/or the updated nomenclature table using the cell set label tag. Finally, cells are mapped to remaining cell sets (if any) using custom scripts. This results in a table of binary calls (0 = no, 1 = yes), indicating exclusion or inclusion of each cell in each cell set (*Figure 1D*), which is written to another csv file as part of the process. This format is designed to allow for probabilistic mapping of cells to cell sets, which is beyond the scope of this manuscript. *These output files are intended to be directly included as supplemental materials in manuscripts performing cell type classification in any species.* In addition, the GitHub repository will be updated to include conversion functions to allow input into future community-accepted cell type databases, as such resources become available. *Supplementary file 1* includes a table of applied nomenclature for all taxonomies discussed in this manuscript, along with cell to cell set mappings for a few example taxonomies.

## Acknowledgements

The authors would like to acknowledge general input and considerations on aspects of cell nomenclature from attendees and affiliates of the workshop, 'Defining an Ontological Framework for a Brain Cell Type Taxonomy: Single-Cell -omics and Data-Driven Nomenclature', held in Seattle, WA, June 2019, including Alex Pollen, Alex Wiltschko, Alexander Diehl, Andrea Beckel-Mitchener, Angela Pisco, Anna Maria Masci, Anna-Kristin Kaufmann, Anton Arkhipov, Aviv Regev, Becky Steck, Bishen Singh, Brad Spiers, Chris Mungall, Christophe Benoist, Cole Trapnell, Dan Geschwind, David Holmes, David Osumi-Sutherland, Davide Risso, Deep Ganguli, Detlev Arendt, Ed Callaway, Eran Mukamel, Evan Macosko, Fenna Krienen, Gerald Quon, Giorgio Ascoli, Gordon Shepherd, Guoping Feng, Hanqing Liu, Jay Shendure, Jens Hjerling-Leffler, Jessica Peterson, Joe Ecker, John Feo, John Marioni, John Ngai, Jonah Cool, Josh Huang, Junhyong Kim, Kelly Street, Kelsey Montgomery, Kara Woo, Lindsay Cowell, Lucy Wang, Luis De La Torre Ubieta, Mark Musen, Maryann Martone, Michele Solis, Ming Zhan, Nicole Vasilevsky, Olga Botvinnik, Olivier Bodenreider, Owen White, Peter Hunter, Peter Kharchenko, Rafael Yuste, Rahul Satija, Richard Scheuermann, Samuel Kerrien, Sean Hill, Sean Mooney, Sten Linnarsson, Tim Jacobs, Tim Tickle, Tom Nowakowski, Uygar Sümbül, Vilas Menon, and Yong Yao. We thank the NIH and CZI for generous co-sponsorship of this workshop. We would also like to acknowledge the many members of the Allen Institute, past and present, who contribute to or support the development of data and analysis of brain cell types - and the organization of this information, particularly Christof Koch, Kimberly Smith, Zizhen Yao, Carol Thompson, Rebecca Hodge, Jonathan Ting, Lucas Graybuck, Thuc Nguyen, Jim Berg, Staci Sorensen, Nik Jorstad, Susan Sunkin, Stefan Mihalas, Rob Young, Tim Fliss, Lydia Ng, Shoaib Mufti, and Stephanie Mok. Research and methods reported in this publication were supported by the Allen Institute, and the National Institute of Mental Health of the National Institutes of Health under award numbers U19MH114830 (to HZ) and U01MH114812 (to ESL). The content is solely the responsibility of the authors and does

not necessarily represent the official views of the National Institutes of Health. The authors would like to thank the Allen Institute founder, Paul G Allen, for his vision, encouragement and support.

## Additional information

### Funding

| Funder | Grant reference number | Author |
|---|---|---|
| Allen Institute | | Jeremy A Miller<br>Nathan W Gouwens<br>Bosiljka Tasic<br>Forrest Collman<br>Cindy TJ van Velthoven<br>Trygve E Bakken<br>Michael J Hawrylycz<br>Hongkui Zeng<br>Ed S Lein<br>Amy Bernard |
| National Institute of Mental Health | U19MH114830 | Hongkui Zeng |
| National Institute of Mental Health | U01MH114812 | Ed S Lein |

The funders had no role in study design, data collection and interpretation, or the decision to submit the work for publication.

### Author contributions

Jeremy A Miller, Conceptualization, Resources, Data curation, Software, Formal analysis, Visualization, Methodology, Writing - original draft, Project administration, Writing - review and editing; Nathan W Gouwens, Conceptualization, Resources, Data curation, Software, Formal analysis, Visualization, Methodology, Writing - original draft, Writing - review and editing; Bosiljka Tasic, Conceptualization, Resources, Funding acquisition, Methodology, Writing - review and editing; Forrest Collman, Cindy TJ van Velthoven, Trygve E Bakken, Conceptualization, Resources, Data curation, Software, Formal analysis, Validation, Visualization, Methodology, Writing - review and editing; Michael J Hawrylycz, Writing - review and editing; Hongkui Zeng, Ed S Lein, Resources, Funding acquisition, Writing - review and editing; Amy Bernard, Conceptualization, Resources, Methodology, Writing - original draft, Project administration, Writing - review and editing

### Author ORCIDs

Jeremy A Miller ⬡ https://orcid.org/0000-0003-4549-588X
Nathan W Gouwens ⬡ https://orcid.org/0000-0001-8429-4090
Bosiljka Tasic ⬡ http://orcid.org/0000-0002-6861-4506
Forrest Collman ⬡ http://orcid.org/0000-0002-0280-7022
Cindy TJ van Velthoven ⬡ http://orcid.org/0000-0001-5120-4546
Trygve E Bakken ⬡ http://orcid.org/0000-0003-3373-7386
Michael J Hawrylycz ⬡ http://orcid.org/0000-0002-5741-8024
Hongkui Zeng ⬡ http://orcid.org/0000-0002-0326-5878
Ed S Lein ⬡ http://orcid.org/0000-0001-9012-6552
Amy Bernard ⬡ https://orcid.org/0000-0003-2540-1153

### Decision letter and Author response

Decision letter https://doi.org/10.7554/eLife.59928.sa1
Author response https://doi.org/10.7554/eLife.59928.sa2

## Additional files

### Supplementary files

• Supplementary file 1. Output files from applying the CCN on 17 taxonomies. This file contains annotated cell sets from all 17 taxonomies shown in *Table 4* along with annotated dendrograms and cell to cell set assignments for a subset of these taxonomies. This file is available on GitHub (https://github.com/AllenInstitute/nomenclature).

• Supplementary file 2. A set of aligned aliase in mammalian M1, reproduced from *Bakken et al., 2020a*. These terms are also applicable to other cortical areas, representing a starting point for future cell type classification efforts and for ontology curation. InterLex identifiers are provided in parentheses when available (*Adkins et al., 2020*).

• Transparent reporting form

### Data availability

This work describes the creation of a cell type nomenclature convention that will, with adoption by the community, become a standard. The data cited is open data available though the Allen Institute web portal, https://brain-map.org. An open Forum is available to engage the community in further development, at https://portal.brain-map.org/explore/classes/nomenclature. Data referenced in this study is also made available according the terms of NIH's Brain Research through Advancing Innovative Neurotechnologies (BRAIN) Initiative - Cell Census Network (BICCN), through the Brain Cell Data Center portal, https://biccn.org/ and https://biccn.org/data.

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
