## [Decision Letter]

**Acceptance summary:**

Defining and naming brain cell types has been a long-standing challenge in neuroscience. New high-throughput techniques have facilitated the generation of many large datasets that detail multi-modal information about cell types. This manuscript presents a system for a generalizable nomenclature that can be applied to the mammalian brain. The system will permit researchers to compare their own cell-type data with other published datasets and facilitate consistent cell-type naming.

**Decision letter after peer review:**

Thank you for submitting your article "Cell type nomenclature for the mammalian brain: Development and application of a systematic, extensible convention" for consideration by *eLife*. Your article has been reviewed by three peer reviewers, including Genevieve Konopka as the Reviewing Editor and Reviewer #1, and the evaluation has been overseen by Catherine Dulac as the Senior Editor. The following individuals involved in review of your submission have agreed to reveal their identity: Fenna Krienen (Reviewer #2); Joseph D Dougherty (Reviewer #3).

The reviewers have discussed the reviews with one another and the Reviewing Editor has drafted this decision to help you prepare a revised submission.

Summary:

All reviewers were in agreement that this paper presents some potential ways of tackling an important problem. However, we think there are some aspects of the paper that could be improved for clarity and to make it accessible to a broad audience. The new policy at *eLife* is to write a summary of essential revisions for the authors when a revised manuscript is warranted and not necessarily send the authors the full reviews.

Our essential revisions include: (1) more details on the immediate use of the system and potentially a step-by-step guide-it definitely seems like this system needs to be much more user friendly even for programmers; (2) how this approach would work without a reference set and ultimately the plan for oversight of such a reference; and (3) how to apply this across species, regions, and modalities.

Even though the new policy is not to send the full reviews, the reviewers each had some useful ideas and nuances about the essential revisions so we are also attaching them here. We do not expect you to address each and every one of these points/comments but rather take them into consideration as you address the essential revisions.

Reviewer #1:

This paper provides an important foundation to develop a universal nomenclature system for vertebrate cell types in single-cell sequencing studies. Similar to efforts to establish gene nomenclature guidelines, this resource is necessary to improve comparisons between datasets and species. As the authors note, any useful system will need to be widely agreed upon and adopted by scientists, and this paper is a good step in initiating that process. I have some general comments on the manuscript and the system that could be implemented. However, I imagine that there might be further modifications that would occur after publication of this manuscript.

1) The glutamatergic nomenclature scheme is neocortical-centric. The "layer" designation should really be a regional specification. For example, "L2/3 IT 4" could have a prefix indicating neocortex like "N L2/3 IT 4." This could allow for expansions including layers in non-neocortical regions, like hippocampus (e.g., "H L2"), or non-layered regions of the pallium like the claustrum (e.g., "C IT"). This would also enhance applicability for non-mammalian brains lacking layered organizations.

2) Glutamatergic cell types across broad vertebrate taxa (e.g. reptiles, birds, mammals) cannot be universally compared to layer-specific nomenclature in mammals because neocortical organization is unique to mammals. Therefore, a universal reference taxonomy must be supplied independently for each major grouping, and homologies can be suggested by, but not defined from, nomenclature. In general, it would be best to emphasize that similarities in cell types are not necessarily indicative of homologies for those cell types. A better description of how one might deal with cell types that diverge across species might be warranted.

3) At some point, the integration of spatial information (other than layers, such as dorsal or ventral) into single-cell sequencing experiments may become commonplace. This nomenclature scheme should be able to incorporate a spatial component if this information proves meaningful, similar to how the authors describe the use of electrophysiological data.

4) The taxonomy ID format CS[YYMMDD][#] is generally fine but note that a four-digit year notation would remove date ambiguities and is in line with universal date formats (ISO 8601).

5) "Cell set label" seems superfluous when "Cell set accession ID" can sufficiently identify each unique cell group. The distinction between identifying "neuron" versus "non-neuron" does not seem to provide enough meaningful information to warrant a separate identity. Table 1 already mentions that this label may be redundant.

6) How/ what steps will be taken to vet the data to include in the taxonomy? As the authors mention there are so many groups generating single cell data but not all of these datasets are of great quality – especially for a naming convention. Will BICCN do this? I imagine the HCA folks might only take care of human data or certain model systems? Who will be in charge of defining the reference cell types?

7) How will the batch differences (lab, sequencing method, machine) be handled?

8) It was mentioned that the same mapping and clustering technique will be applied for all datasets. -Is it possible that certain datasets might be more amenable to "tweaks" in a pipeline rather than a one size fits all approach?

Reviewer #2:

This manuscript presents a proposal for a generalizable cell type nomenclature convention system for the mammalian brain. How to define and name brain cell types is a longstanding issue; there is currently no standard convention. The recent explosion of large, single cell datasets based on molecular measurements (RNA, epigenetics) means there is both an opportunity to gain detailed and often multi-modal information about cell types, as well as a need to reconcile nomenclatures across studies. As such, this article presents a thoughtful, implementable system for a standardized nomenclature, as well as a discussion of some infrastructure and governance considerations that would facilitate widespread community adoption.

1) I read this project as having two components: (A) an immediately implementable nomenclature system (with associated code for end-users to run on their own data), and (B) a proposal outlining possible infrastructures that would support 'alignment' of community data to reference(s) and linking cell type information across studies (including the need for computational infrastructure, governance). I think there should be better motivation for why (A) should be adopted without (B). The authors state that the schema will be "immediately useful" but an end-user may not see the practical advantage over their own in-house conventions (unless/until there is a reference, governance, ontology with controlled values, etc).

2) Practically speaking, the article (or at least the github repo) should also clearly state what best-practices adoption of (A) would look like before a data repository is established, i.e. which outputs or terms are used in figures and tables, should full outputs be included as supplementary data (spreadsheets) in manuscripts etc. The schema introduces quite a few new terms and conventions and I think you have to be more explicit for how end-users should incorporate it in their own work. This could be achieved by more detailed examples as well as guidance in the github (note I ran the code using the supplied MTG dataset but did not try applying it to new data. Authors might consider adding vignettes that take in common algorithm outputs, e.g. output from Seurat, 10X cell ranger pipeline etc).

3) Multiple examples of applying the nomenclature schema to published datasets are given. I wonder whether it would be more effective to focus on one dataset. Figure 1 (human transcriptomic MTG data) and Figure 2 (mouse transcriptomic VISp data) largely do the same work, though they are displayed and formatted differently, which is a bit confusing. Figure 5 presents an example of creating a reference from the data in Figures 1 and 2, but several aspects are not clear: in (B), how are the preferred aliases named, (C) does "Human types" = cell set labels in Figure 1B, (D) how are the lines that visually link the modalities derived, and how are they formally represented in the nomenclature system. The final example (multi-modal, multi species comparison of cortical area M1) is also complex as it contains multiple datatypes as well as a derived 'reference', but as currently presented is not very effective in showing how the nomenclature is applied or how taxonomies are linked. I wonder whether it would be more effective to focus either on the taxonomies presented in Figures 1, 2 and 5, or alternatively on this large M1 study, and fully unpack how to apply and visually represent the schema with just one of these. Alternatively, one could start with toy examples that illustrate the process before applying to real data (again one might be better than several).

4) Creating or using a reference is not explicitly part of the proposed nomenclature schema, but clearly has great utility in terms of linking taxonomies (Figure 5 and 6). The authors could consider describing in more detail the considerations they have made in forming such cross dataset or cross species references. How do user-generated references fit in the proposed system – can the same classification system be applied (i.e. each reference has a taxonomy ID, each cluster has a cell set accession ID) or are there other metadata that should be included? Authors might consider a separate figure devoted to applying the nomenclature schema to a derived reference (e.g. unpacking something like Figure 5B).

5) If I understand correctly, cell set aliases can be based on seemingly very different types of evidence, including (1) quantitative alignment to a reference, (2) user inference based on observation of shared features such as marker genes (i.e., both datasets have Chodl+ cluster, so it is probably the same), and (3) inference based on prior knowledge (an ME cell set that has a location and electrophysiological profile consistent with chandelier cells is aliased to a transcriptomic cell set named "chandelier"). These are really different types of evidence and perhaps should be controlled or distinguished in the system.

Reviewer #3:

This paper is a thoughtful contribution to a tough problem and represents a reasonable step in the right direction. I think it would fit well with *eLife* and form the basis for beginning of better cross-paper curation of scRNAseq data and other related datasets. It is not the full solution, but is careful in its claims and I think will be an important part of the conversation towards those larger solutions. I have some moderate recommendations for revisions.

1) For cell set accession IDs, they may want to include a '.' between the CS191012 and the unique number for the cell set, and then just iterate the numbers up (.1, .2, .3, … .12 … .10000). The current scheme will max out at 1000 cell sets. That might seem like a huge number now, but someone soon will do 10x on the whole body in one paper and need more than 1000 cell sets.

2) I don't have the bandwidth at the moment to do this as a reviewer, but I would recommend they consider approaching 2-3 labs outside of their group (i.e. external Beta testers) and have them try to enter on of their datasets into this structure using the GitHub code and see how it goes, using only this paper and the associated materials as instructions and iron out any wrinkles or misunderstandings that emerge. If you want this to roll out smoothly, you want researchers' first experiences trying it to be positive to help promote wide adoption.

3) I would recommend adding a section (or perhaps a supplement) that is a clear checklist of what to do as an end researcher who might want to adopt this. If you've convinced me to do this with my data, what are the explicit and actionable recommendations for what I should do? Is this meant to be like submitting your data to GEO? Where any paper publishing a scRNAseq dataset will adopt this standardized approach to naming as the simultaneously upload their taxonomy to a particular database in a standardized format, and put a link in their Materials and methods section? I feel like this is not quite proposing that (as no such database was highlighted, though they highlight the need for one). Or rather is the hope that anyone who generates a scRNAseq dataset will provide their taxonomy in a standard file format as a supplement to their paper? If so, defining a file type (a .txon file or something?) that you are recommending everyone generate and add as supplement might be what you are championing. Explicitly naming that filetype(s) and making that recommendation might help. (If so, is that something that could be rolled into standard analysis packages – e.g. Seuret? That would lower the barrier to adoption) Or is this more like just trying to have everyone agree to use standardized gene names when they mention them? But not necessarily provide supplementary files. Like just being careful how you format your writing and figures like how I should use Pvalb for mouse genes and PVALB for human and PVALB for the protein, and not PV, PVA, etc? Anyway, I just wanted more concrete recommendations of what our expectations as authors (and as reviewers) ought to be for adoption of this standard.

As a starting point, perhaps just recommending a defined file type generated by the code (the .txon file) be included as a supplement is a reasonable recommendation at this time.

Basically, overall I think this paper is making an important and timely contribution. It did a good job of explaining their solution to addressing some of the challenges for annotating these datasets, but stopped just short of a concrete guide on how one could implement it in the near term.

---

## [Author Response]

Summary:All reviewers were in agreement that this paper presents some potential ways of tackling an important problem. However, we think there are some aspects of the paper that could be improved for clarity and to make it accessible to a broad audience. The new policy at eLife is to write a summary of essential revisions for the authors when a revised manuscript is warranted and not necessarily send the authors the full reviews.Our essential revisions include: (1) more details on the immediate use of the system and potentially a step-by-step guide-it definitely seems like this system needs to be much more user friendly even for programmers; (2) how this approach would work without a reference set and ultimately the plan for oversight of such a reference; and (3) how to apply this across species, regions, and modalities.

We have reorganized our manuscript to include sections addressing each of these requests. The section titled, “Step by step application of the CCN to human MTG,” walks through the process of applying the CCN to an example data set in human middle temporal gyrus using the GitHub code. In addition, the GitHub repository itself has improved annotation, so that it is accessible as an application independent of this manuscript. After the proposal of a cell typing convention for the mammalian cortex, we present the concept of a reference atlas, and complementarity to the CCN. While there is nothing in the schema itself distinguishing a reference taxonomy from any other taxonomy, the reference taxonomies are primarily used to define aligned aliases and to facilitate alignment of other data sets. We present a specific example reference taxonomy created from the primary motor cortex (M1) in mouse, marmoset, and human, which replaces the human MTG / mouse VISp taxonomy presented in the initial submission. Governance of such a reference is beyond the scope of this manuscript, and we have substantially cut the sections related to governance and potential for future work for this reason. That being said, we provide some evidence for why our current reference is a good reference and present a collection of 18 taxonomies that use this integrated taxonomy as their reference framework.

Finally, we added a section, “Applying the CCN to existing and new datasets,” which presents five examples for how the CCN can be used right now, in the absence of a centralized repository for storing taxonomies and cell sets. This section reframes some of the use cases presented in the initial submission, and also includes some new use cases that together address how to “apply this across species, regions, and modalities.” For example, there are use cases on using the CCN to apply nomenclature to a non-mammalian brain structure, the turtle pallium, in the context of the presented reference taxonomy; and, adding aligned aliases to human MTG based on alignment with human M1; and, on mapping a taxonomy defined using cell morphology and biophysics to the cell type reference taxonomy, based on transgenically defined cell markers. The final two use cases explore biologically relevant inferences about a specific cell type (*Sst-Chodl*) based on aligned aliases, and on comparison of a novel data set to a set of 18 compiled taxonomies. Together, we feel that these changes address the reviewer concerns, leading to much stronger manuscript and nomenclature schema overall.

Even though the new policy is not to send the full reviews, the reviewers each had some useful ideas and nuances about the essential revisions so we are also attaching them here. We do not expect you to address each and every one of these points/comments but rather take them into consideration as you address the essential revisions.Reviewer #1:This paper provides an important foundation to develop a universal nomenclature system for vertebrate cell types in single-cell sequencing studies. Similar to efforts to establish gene nomenclature guidelines, this resource is necessary to improve comparisons between datasets and species. As the authors note, any useful system will need to be widely agreed upon and adopted by scientists, and this paper is a good step in initiating that process. I have some general comments on the manuscript and the system that could be implemented. However, I imagine that there might be further modifications that would occur after publication of this manuscript.1) The glutamatergic nomenclature scheme is neocortical-centric. The "layer" designation should really be a regional specification. For example, "L2/3 IT 4" could have a prefix indicating neocortex like "N L2/3 IT 4." This could allow for expansions including layers in non-neocortical regions, like hippocampus (e.g., "H L2"), or non-layered regions of the pallium like the claustrum (e.g., "C IT"). This would also enhance applicability for non-mammalian brains lacking layered organizations.

This is an excellent point and was actually a primary focus of the workshop, “Defining an Ontological Framework for a Brain Cell Type Taxonomy: Single-Cell -omics and Data-Driven Nomenclature,” held in Seattle WA, June 2019, and from which many ideas is this manuscript originated. Unfortunately, there was very little consensus at this workshop on specifically what we should call these cell types and what level of “regional” resolution (e.g., “middle temporal gyrus” vs. “neocortex” vs. “brain”) should be used. For example, some cell types are likely region-specific (e.g., glutamatergic types, as mentioned) while others (e.g., glia) likely span multiple regions, and in many cases the regional resolution may change depending on the algorithms used for cell typing. In this case, we have attempted to bypass this issue somewhat by dividing the CCN into a schema component (which works with any taxonomy) and a specific naming convention that is limited to mammalian neocortex. Within neocortex, we further address this by implementing a proposed solution from the workshop: to define two distinct ontologies for each cell set. In additional to the cell type ontology that would be mostly region-agnostic, we provide two slots to assign a specific brain (or body) structure (“cell_set_structure”) and an associated tag a cross-species structural ontology (“cell_set_ontology_tag”) such as UBERON or a single-species ontology (e.g. from http://atlas.brain-map.org/) if higher resolution cell types are needed.

2) Glutamatergic cell types across broad vertebrate taxa (e.g. reptiles, birds, mammals) cannot be universally compared to layer-specific nomenclature in mammals because neocortical organization is unique to mammals. Therefore, a universal reference taxonomy must be supplied independently for each major grouping, and homologies can be suggested by, but not defined from, nomenclature. In general, it would be best to emphasize that similarities in cell types are not necessarily indicative of homologies for those cell types. A better description of how one might deal with cell types that diverge across species might be warranted.

The intention of this nomenclature schema was to provide guidance for typing of cells in the mammalian cortex, as indicated in the title of our manuscript. One of the primary reasons for restricting our naming conventions to mammals is for the precise reason mentioned: that they may break down in other species. However, the schema itself is agnostic to species (or organ system) and can be run independently of a universal reference taxonomy or can still be linked to such a taxonomy using the aligned, preferred, and additional alias tags. For example, we now present as a use case the example of a turtle pallium which was aligned with mouse neocortex (to the extent possible) in the initial publication (Tosches et al., 2018). In this case, non-neuronal cells and GABAergic interneurons could be matched to some degree between species, and this information was captured by assigning aligned aliases of Pvalb, Sst, OPC, microglia, etc. to the appropriate cell sets in turtles. For glutamatergic neurons, only superficial vs. deep layer neurons could be distinguished, and these show an inversion between clades. This information is captured by the additional alias tag, and this entire taxonomy with applied CCN is included as part of Additional File 1 in our updated submission. Finally, it is worth reiterating that data-driven strategies for linking (or failing to link) such cross-species taxonomies are assumed to have already occurred prior to applying the CCN and that the goal of the CCN is to capture these existing links. It is also worth noting that no links to a reference taxonomy are required for application of the CCN to be of some use. We have updated the text of the manuscript to address these points.

3) At some point, the integration of spatial information (other than layers, such as dorsal or ventral) into single-cell sequencing experiments may become commonplace. This nomenclature scheme should be able to incorporate a spatial component if this information proves meaningful, similar to how the authors describe the use of electrophysiological data.

As mentioned in #1 above, we now add a “cell set structure” tag and an associated “cell set ontology tag” to our nomenclature schema, which should address integration of spatial information.

4) The taxonomy ID format CS[YYMMDD][#] is generally fine but note that a four-digit year notation would remove date ambiguities and is in line with universal date formats (ISO 8601).

Our taxonomy ID was intended to strike the balance between a long but highly informative name (e.g., “MTG-20201020-Transcriptomics”) and a short generic name (e.g., “t251”), while ensuring that all taxonomies remain unique in the short term. However, we recognize this weakness, along with the weakness of having a maximum of 1000 cell sets (as mentioned below). Given the reviewer feedback, we have made the following changes, which are now implemented in the manuscript and associated CCN scripts:

– Taxonomy_id = CCN[YYYYMMDD]#

– Cell_set_id = CS[YYYYMMDD]#_#

5) "Cell set label" seems superfluous when "Cell set accession ID" can sufficiently identify each unique cell group. The distinction between identifying "neuron" versus "non-neuron" does not seem to provide enough meaningful information to warrant a separate identity. Table 1 already mentions that this label may be redundant.

As multiple reviewers (and co-authors) agree with this assessment, we have dramatically reduced discussion of cell_set_label from the nomenclature schema in this version of the manuscript. The tag itself is important for the GitHub scripts to work properly and so the term remains, but it is only discussed in this context.

6) How/ what steps will be taken to vet the data to include in the taxonomy? As the authors mention there are so many groups generating single cell data but not all of these datasets are of great quality – especially for a naming convention. Will BICCN do this? I imagine the HCA folks might only take care of human data or certain model systems? Who will be in charge of defining the reference cell types?

The advantage of this nomenclature schema is that it could be run on any taxonomy, regardless of data modality or data quality. For example, a small lab could perform a clustering and integration analysis (e.g., with a pre-existing reference), and then name cell types using this schema in such a way that cell sets are linked with the reference. In this case, the reference would not be changed and the quality of the single cell data does not matter. The issue of governance is a challenging one that is beyond the scope of this manuscript, and therefore we have reduced our discussion and speculation on this topic. That being said, we feel that the new reference taxonomy presented here (cross-species primary motor cortex) represents a good starting reference taxonomy for mammalian cortex because it represents a well-curated taxonomy presented as the output of a multi-lab collaboration through the BICCN. To aid in its use we present all a collection of 18 taxonomies which use the aligned alias terms introduced in this taxonomy as Additional File 1.

7) How will the batch differences (lab, sequencing method, machine) be handled?

Batch differences should be handed prior to application of any nomenclature schema. Strategies for doing so are beyond the scope of this manuscript.

8) It was mentioned that the same mapping and clustering technique will be applied for all datasets. -Is it possible that certain datasets might be more amenable to "tweaks" in a pipeline rather than a one size fits all approach?

Clustering and alignment should be handed prior to application of any nomenclature schema. Strategies for doing so are beyond the scope of this manuscript.

Reviewer #2:[…] 1) I read this project as having two components: (A) an immediately implementable nomenclature system (with associated code for end-users to run on their own data), and (B) a proposal outlining possible infrastructures that would support 'alignment' of community data to reference(s) and linking cell type information across studies (including the need for computational infrastructure, governance). I think there should be better motivation for why (A) should be adopted without (B). The authors state that the schema will be "immediately useful" but an end-user may not see the practical advantage over their own in-house conventions (unless/until there is a reference, governance, ontology with controlled values, etc).

We view (A) as containing two distinct components which are applicable in different (but overlapping) use cases, but otherwise agree with this assessment. First, the immediately implementable nomenclature system revolving around the idea of cell sets and taxonomies. Second, a specific convention for naming mammalian neocortical cell types, which presents a starting proposal for a common language to allow linking of cell types between studies. With this in mind, we have attempted to improve on our explanation for why (A) should be adopted without (B). This can be found in the section titled “Applying the CCN to existing and new datasets”, which includes five use cases for immediate application of the schema. Of particular note, we now present a series of taxonomies (including a reference taxonomy) on which the CCN has already been applied and to which other groups can immediately compare their results (see use case #5 and Supplementary file 1). This being said, we recognize that having an ontology with controlled values, a centralized database, and governance could increase utility and are actively pursuing this line of work. Such topics are challenging and beyond the scope of the manuscript and we therefore have reduced our discussion and speculation on these topics.

2) Practically speaking, the article (or at least the github repo) should also clearly state what best-practices adoption of (A) would look like before a data repository is established, i.e. which outputs or terms are used in figures and tables, should full outputs be included as supplementary data (spreadsheets) in manuscripts etc. The schema introduces quite a few new terms and conventions and I think you have to be more explicit for how end-users should incorporate it in their own work. This could be achieved by more detailed examples as well as guidance in the github (note I ran the code using the supplied MTG dataset but did not try applying it to new data. Authors might consider adding vignettes that take in common algorithm outputs, e.g. output from Seurat, 10X cell ranger pipeline etc).

This is a great suggestion. Our goal for this manuscript is to have a system that could be implemented by a computational or non-computational researcher by following step-by-step instructions. To this end the GitHub repo has been updated with more complete step-by-step instructions for implementation (along with corresponding code), and these instructions have been described in the manuscript in more detail than in the initial analysis. We have also made some edits to the CCN (described below) and have changed the GitHub code accordingly. As part of these updates, we have included text regarding what to do after implementation as related to proposed best practices for publication with the goal of allowing future intake into a centralized database directly from manuscript Supplementary files. The idea of providing vignettes for applying the CCN as part of a Seurat or 10x cell ranger pipeline is a great one that we hope to revisit at a later time.

3) Multiple examples of applying the nomenclature schema to published datasets are given. I wonder whether it would be more effective to focus on one dataset. Figure 1 (human transcriptomic MTG data) and Figure 2 (mouse transcriptomic VISp data) largely do the same work, though they are displayed and formatted differently, which is a bit confusing. Figure 5 presents an example of creating a reference from the data in Figures 1 and 2, but several aspects are not clear: in (B), how are the preferred aliases named, (C) does "Human types" = cell set labels in Figure 1B, (D) how are the lines that visually link the modalities derived, and how are they formally represented in the nomenclature system. The final example (multi-modal, multi species comparison of cortical area M1) is also complex as it contains multiple datatypes as well as a derived 'reference', but as currently presented is not very effective in showing how the nomenclature is applied or how taxonomies are linked. I wonder whether it would be more effective to focus either on the taxonomies presented in Figure 1, 2 and 5, or alternatively on this large M1 study, and fully unpack how to apply and visually represent the schema with just one of these. Alternatively, one could start with toy examples that illustrate the process before applying to real data (again one might be better than several).

Our initial strategy was to present a number of different examples for use of this nomenclature schema, but the point about presenting a single application in more depth and focus is well taken. To address this comment we have reorganized the manuscript as described in the response to the next comment, including redefining the multi-species M1 taxonomy as the reference and removing the figures on mouse VISp (although the taxonomies are still retained in Supplementary file 1).

4) Creating or using a reference is not explicitly part of the proposed nomenclature schema, but clearly has great utility in terms of linking taxonomies (Figures 5 and 6). The authors could consider describing in more detail the considerations they have made in forming such cross dataset or cross species references. How do user-generated references fit in the proposed system – can the same classification system be applied (i.e. each reference has a taxonomy ID, each cluster has a cell set accession ID) or are there other metadata that should be included? Authors might consider a separate figure devoted to applying the nomenclature schema to a derived reference (e.g. unpacking something like Figure 5B).

Reviewer #1 had similar concerns, which are addressed above. In summary, the advantage of the CCN is that it could be run on any taxonomy, regardless of data modality, data quality, or origin species (although some modifications to the specific naming conventions will likely be needed for distant species and for structures outside of neocortex). For example, a small lab could perform a clustering and integration analysis (e.g., with a pre-existing reference), and then name cell types using this system in such a way that cell sets are linked with the reference. In addition, reference taxonomies can be treated identically to any other taxonomy in this schema, with the caveat that aligned aliases are more likely to be generated through reference taxonomies and through other taxonomies (although this is not required). We have reorganized the structure of our paper to (1) present an example of a single taxonomy run by itself (human MTG) with direct tie-in to the GitHub repo, (2) present the idea of a reference taxonomy along with an example from the primary motor cortex, and then (3) present the same human MTG example but in the context of this M1 reference taxonomy. In principle, this also addresses the issue of immediate utility as all of this is done without a governing body, and is presented as the first such use case. We don’t explicitly address the topic of metadata in the manuscript, but there are likely to be other pieces of meta-data that may be particularly relevant to different taxonomies (e.g., species). These can be provided either as separate meta-data tables or as optional additional columns in the CCN.

5) If I understand correctly, cell set aliases can be based on seemingly very different types of evidence, including (1) quantitative alignment to a reference, (2) user inference based on observation of shared features such as marker genes (i.e., both datasets have Chodl+ cluster, so it is probably the same), and (3) inference based on prior knowledge (an ME cell set that has a location and electrophysiological profile consistent with chandelier cells is aliased to a transcriptomic cell set named "chandelier"). These are really different types of evidence and perhaps should be controlled or distinguished in the system.

This is a good point. To address this, we have added tags for “cell_set_alias_assignee” and “cell_set_alias_citation” to the CCN as a mechanism for providing both credit and evidence for alias terms. While this does not distinguish different types of evidence, it does provide both a person to ask and a place to look for such evidence, and if there are multiple aliases, there can also be more than one assignee or citation. We feel that this is a good starting point at addressing evidence but are certainly open to suggestions for ways of more carefully capturing this point without increasing the complexity of the CCN to a great extent.

Reviewer #3:This paper is a thoughtful contribution to a tough problem and represents a reasonable step in the right direction. I think it would fit well with eLife and form the basis for beginning of better cross-paper curation of scRNAseq data and other related datasets. It is not the full solution, but is careful in its claims and I think will be an important part of the conversation towards those larger solutions. I have some moderate recommendations for revisions.1) For cell set accession IDs, they may want to include a '.' between the CS191012 and the unique number for the cell set, and then just iterate the numbers up (.1, .2, .3, … .12 … .10000). The current scheme will max out at 1000 cell sets. That might seem like a huge number now, but someone soon will do 10x on the whole body in one paper and need more than 1000 cell sets.

Reviewer 1 had related feedback and given feedback from both reviewers, we have made the following changes:

– Taxonomy_id = CCN[YYYYMMDD]#

– Cell_set_id = CS[YYYYMMDD]#_#

2) I don't have the bandwidth at the moment to do this as a reviewer, but I would recommend they consider approaching 2-3 labs outside of their group (i.e. external Beta testers) and have them try to enter on of their datasets into this structure using the GitHub code and see how it goes, using only this paper and the associated materials as instructions and iron out any wrinkles or misunderstandings that emerge. If you want this to roll out smoothly, you want researchers' first experiences trying it to be positive to help promote wide adoption.

This is a great suggestion and we are actively pursuing related collaborations. At the moment our collaborations are focused more on other consortia, where we are working with representatives from HubMAP and HCA to at least make sure that this system is compatible with similar systems being designed there. In particular, Peter Kharchenko’s group, who is developing the Cell Annotation Platform, has been providing direct feedback on the GitHub repo by running test cases. This specific suggestion of approaching external groups to Beta test is excellent and will be pursued in the near future, but is beyond the scope of the manuscript.

3) I would recommend adding a section (or perhaps a supplement) that is a clear checklist of what to do as an end researcher who might want to adopt this. If you've convinced me to do this with my data, what are the explicit and actionable recommendations for what I should do? Is this meant to be like submitting your data to GEO? Where any paper publishing a scRNAseq dataset will adopt this standardized approach to naming as the simultaneously upload their taxonomy to a particular database in a standardized format, and put a link in their Materials and methods section? I feel like this is not quite proposing that (as no such database was highlighted, though they highlight the need for one). Or rather is the hope that anyone who generates a scRNAseq dataset will provide their taxonomy in a standard file format as a supplement to their paper? If so, defining a file type (a .txon file or something?) that you are recommending everyone generate and add as supplement might be what you are championing. Explicitly naming that filetype(s) and making that recommendation might help. (If so, is that something that could be rolled into standard analysis packages – e.g. Seuret? That would lower the barrier to adoption) Or is this more like just trying to have everyone agree to use standardized gene names when they mention them? But not necessarily provide supplementary files. Like just being careful how you format your writing and figures like how I should use Pvalb for mouse genes and PVALB for human and PVALB for the protein, and not PV, PVA, etc? Anyway, I just wanted more concrete recommendations of what our expectations as authors (and as reviewers) ought to be for adoption of this standard.As a starting point, perhaps just recommending a defined file type generated by the code (the .txon file) be included as a supplement is a reasonable recommendation at this time.

This comment is a lot to unpack, but I think is mostly addressed in responses to reviewers 1 and 2, who had similar suggestions for adjustments to the manuscript. In particular, the GitHub repo has been updated with more complete step-by-step instructions for implementation (along with corresponding code), and these instructions have been described in the manuscript in more detail than in the initial submission. As part of this, we have included steps regarding best practices for publication with the goal of allowing *future* intake into a centralized database directly from manuscript supplementary files. We have also reorganized the paper in a way that should make the relationships with reference taxonomies – both now and after the formation of a governing body--more clear. While we like the idea of having something that could be rolled into standard analysis packages, there is an important manual step related to annotating aligned aliases and (optionally) adding cell sets for cell types in non-hierarchical organization that would make such automation challenging. That being said, we hope that our improved readme with the GitHub repo makes implementation cleaner (and we will test this – see comment #2). Finally, we now propose including a specific file as Supplementary file 1; this is a zip file containing (1) the CCN annotation table for cell sets, (2) cell to cell set mappings, and (3) an updated dendrogram (optional).

Basically, overall I think this paper is making an important and timely contribution. It did a good job of explaining their solution to addressing some of the challenges for annotating these datasets, but stopped just short of a concrete guide on how one could implement it in the near term.

Both other reviewers agree with this assessment, and as a step in this direction we have reorganized the structure of our paper as follows: (1) we present an example of a single taxonomy run by itself (human MTG) with direct tie-in to the GitHub repo, (2) we present the idea of a reference taxonomy along with an example from the primary motor cortex, (3) we present the same human MTG example but in the context of this M1 reference taxonomy as an example of immediate utility, and (4) present four additional use cases that can be done now (e.g., without a governing body or centralized database). We also clarify some aspects related to cross-species, cross-modal, and cross-region nomenclature.